# Localize-and-Stitch: Efficient Model Merging via Sparse Task Arithmetic

**Yifei He**                                                        *yifeihe3@illinois.edu*
*University of Illinois Urbana-Champaign*

**Yuzheng Hu**                                                        *yh46@illinois.edu*
*University of Illinois Urbana-Champaign*

**Yong Lin**                                                        *yl7690@princeton.edu*
*Princeton University*

**Tong Zhang**                                                        *tozhang@illinois.edu*
*University of Illinois Urbana-Champaign*

**Han Zhao**                                                        *hanzhao@illinois.edu*
*University of Illinois Urbana-Champaign*

**Reviewed on OpenReview:** *https://openreview.net/forum?id=9CWU8Oi86d*

## Abstract

Model merging offers an effective strategy to combine the strengths of multiple finetuned models into a unified model that preserves the specialized capabilities of each. Existing methods merge models in a *global* manner, performing arithmetic operations across *all* model parameters. However, such global merging often leads to task interference, degrading the performance of the merged model. In this work, we introduce `Localize-and-Stitch`, a novel approach that merges models in a *localized* way. Our algorithm works in two steps: i) Localization: identify tiny (1% of the total parameters) localized regions in the finetuned models containing essential skills for the downstream tasks, and ii) Stitching: reintegrate only these essential regions back into the pretrained model for task synergy. We demonstrate that our approach effectively locates sparse regions responsible for finetuned performance, and the localized regions could be treated as compact and interpretable representations of the finetuned models (tasks). Empirically, we evaluate our method on various vision and language benchmarks, showing that it outperforms existing model merging methods under different data availability scenarios. Beyond strong empirical performance, our algorithm also facilitates model compression and preserves pretrained knowledge, enabling flexible and continual skill composition from multiple finetuned models with minimal storage and computational overhead. Our code is available at `https://github.com/uiuctml/Localize-and-Stitch`.

## 1 Introduction

Pretrained models (Devlin et al., 2018; Liu et al., 2019; Raffel et al., 2020; Radford et al., 2021) contain a wealth of rich and generalizable information, and finetuning these models for specific downstream tasks significantly enhances performance compared to training from scratch (Chen et al., 2020b). With the growing popularity of the pretrain-finetune paradigm, a vast array of finetuned models have been made available on platforms like Hugging Face (Wolf et al., 2020), and many of them originate from the same pretrained models, such as CLIP (Radford et al., 2021). However, deploying multiple finetuned models independently, each for a different downstream task, incurs large storage and maintenance cost, and limits knowledge transfer across them.

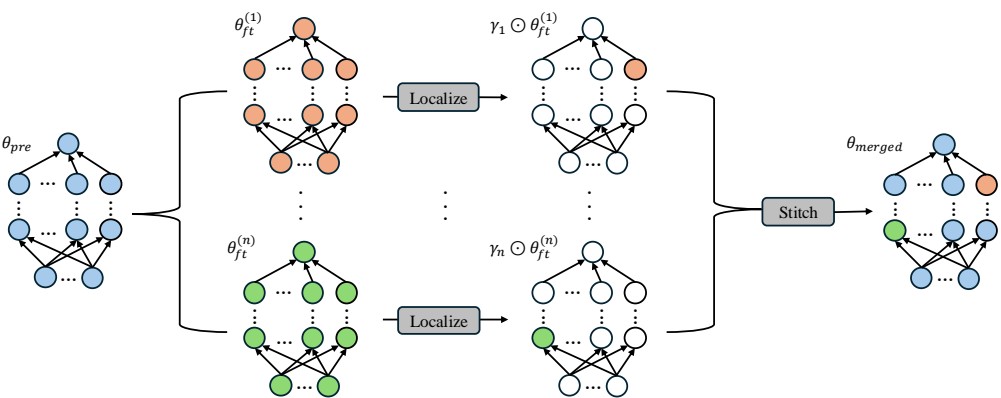

Figure 1: `Localize-and-Stitch`: Given $n$ models $\{\theta_{\text{ft}}^{(i)}\}_{i=1}^{n}$ finetuned from $\theta_{\text{pre}}$, we first localize regions containing skills acquired during finetuning through per-model binary masks $\{\gamma_i\}_{i=1}^{n}$, then stitch the localized regions $\{\gamma_i \odot \theta_{\text{ft}}^{(i)}\}_{i=1}^{n}$ onto the pretrained model, where $\odot$ is the element-wise product. Empty nodes after the localization step mean that the mask is not activated at that position. Since the localized regions are tiny ($\sim 1\%$), we reduce potential task conflicts and make minimal changes to the pretrained model.

Model merging offers a viable solution to these challenges by integrating the strengths of multiple finetuned models into a single model that retains the specialized capabilities of each. The key advantage of model merging over traditional multi-task learning (MTL) (Caruana, 1997; Zhang & Yang, 2021; Hu et al., 2024; He et al., 2024) is its efficiency, in that it does not require joint training on data across all tasks, but only involves arithmetic operations in the weight space. Existing methods merge models by averaging model parameters via arithmetic mean (Wortsman et al., 2022a; Ilharco et al., 2023), Fisher information (Matena & Raffel, 2022), regression mean (Jin et al., 2022) or learned merging weights (Yang et al., 2023). Those methods all average the models in a *global* manner, meaning that they perform arithmetic operations to *all* parameters of the finetuned models. However, similar to the conflicting gradient problem in MTL (Yu et al., 2020; Liu et al., 2021), parameters in different finetuned models often have interference with each other, leading to suboptimal performance of the merged model. Recent works find that redundant parameter updates in finetuning are sources of conflicts (Yadav et al., 2023). Although the majority of model parameters are updated during finetuning, only very few contribute to improving the performance on downstream tasks (Chen et al., 2020a; Hoefler et al., 2021).

To overcome these limitations, we propose `Localize-and-Stitch`, an efficient algorithm that merges models in a *localized* manner. The algorithm (Figure 1) involves two steps: **i) Localization**: identify tiny localized regions in the finetuned models containing essential skills for the downstream tasks. **ii) Stitching**: reintegrate only these essential regions back into the pretrained model. In the experiments, we verify that the changes in finetuend parameters are highly redundant, as we can efficiently identify just 1% of the total parameters that recovers over 99% of the finetuned performance. We evaluate our method on various language and vision tasks, showing that it outperforms existing model merging methods under different data availability scenarios.

Beyond the superior performance on model merging, our approach has several distinct advantages: **i) Interpretability of task relations**: each localized region encapsulates task-specific skills from the finetuned models, and overlap among them is indicative of knowledge sharing. **ii) Model compression**: Our localization method enables compact representation of finetuned models, significantly reducing the storage space to only 1% of the original without sacrificing performance. This enables flexible integration of finetuned models' capabilities with minimal storage and computational overhead. **iii) Preservation of pretrained knowledge**: By making minimal and localized changes to the pretrained model, our merged model maintains its generalizability and achieves superior multi-task performance, effectively mitigating catastrophic forgetting associated with finetuning.

## 2 Preliminaries

**Notation.** Given a set of $n$ tasks, we denote the pretrained model parameters as $\theta_{\text{pre}} \in \mathbb{R}^d$, the model parameters finetuned on the $i$-th task as $\theta_{\text{ft}}^{(i)} \in \mathbb{R}^d$. Note that all $\theta_{\text{ft}}^{(i)}$ are finetuned from the same pretrained model.

Table 1: Baselines for model merging.

| Category | Method | Mathematical expression | Note |
|---|---|---|---|
| Global | Simple averaging (Wortsman et al., 2022a) | $\theta_{\text{merged}} = \frac{1}{n}\sum_{i=1}^{n}\theta_{\text{ft}}^{(i)}$ | Element-wise mean |
| | Task arithmetic (TA) (Ilharco et al., 2023) | $\theta_{\text{merged}} = \theta_{\text{pre}} + \alpha\sum_{i=1}^{n}\tau_i$ | $\alpha$ tuned on a validation set |
| | Fisher merging (Matena & Raffel, 2022) | $\theta_{\text{merged}} = \sum_{i=1}^{n}\hat{F}_i\theta_{\text{ft}}^{(i)} / \sum_{i=1}^{n}\hat{F}_i$ | Weighted by Fisher information matrices |
| | RegMean (Jin et al., 2022) | $\theta_{\text{merged}} = (\sum_{i=1}^{n}X_i^{\top}X_i)^{-1}\sum_{i=1}^{n}(X_i^{\top}X_i\theta_{\text{ft}}^{(i)})$ | Minimizes difference in merged and individual activations |
| | AdaMerging (Yang et al., 2023) | $\{\theta_{\text{merged}}^{l}\}_{l=1}^{L} = \{\theta_{\text{pre}} + \sum_{i=1}^{n}\lambda_i^l\theta_{\text{ft}}^{(i)^l}\}_{l=1}^{L}$ | Layer-wise weights learned by entropy minimization |
| Localized | TIES-Merging (Yadav et al., 2023) | Trims the parameters in task vectors with small magnitudes, elect a sign at each position of the task vector and only keep the parameters with the same sign. | |
| | Consensus TA/TIES (Wang et al., 2024b) | Compute multi-task task vectors: $\tau_{MTL} = \theta_{\text{merged}} - \theta_{\text{pre}}$ with $\theta_{\text{merged}}$ obtained by any merging method. Construct task masks: $m_t = \mathbb{1}\{|\tau_t| \geq |\tau_{MTL} - \tau_t| \cdot \lambda_t\}$. Apply consensus mask $m_{consensus} = \mathbb{1}\{\sum_{t\in[T]}m_t \geq 2\}$ on $\tau_{MTL}$. | |

**Task vectors.** A task vector is the element-wise difference of the finetuned and pretrained parameters, denoted as $\tau_i = \theta_{\text{ft}}^{(i)} - \theta_{\text{pre}} \in \mathbb{R}^d$. These vectors encapsulate the knowledge acquired during the finetuning process. This knowledge can be effectively manipulated through task arithmetic (Ilharco et al., 2023), which involves performing arithmetic operations on task vectors to compose learned skills across tasks.

**Model merging.** The goal of model merging is to efficiently aggregate the parameters of the $n$ finetuned models into a single multi-task model $\theta_{\text{merged}}$ without the need to retrain the model on the initial task-specific data. The resulting merged model should perform well on all the tasks simultaneously.

Existing methods perform merging in the general form $\theta_{\text{merged}} = \theta_{\text{pre}} + \sum_{i=1}^{n}\alpha_i\tau_i$, and their difference mainly lies in the way of determining the scaling factors $\alpha_i$. We introduce the baselines in Table 1, and categorize them into *global* and *localized* methods based on whether the algorithm incorporates selection strategies to identify which parameters to merge. Localized algorithms specifically target sparse and localized regions, while global algorithms merge parameters indiscriminately. We provide detailed comparisons with the two localized algorithms, in Appendix C and Appendix D respectively. Note that AdaMerging has two variants: one learns layer-wise weights and another learns task-wise weights. In this work, we refer to AdaMerging as the layer-wise version because of its superior performance over its task-wise counterpart.

**Data requirements.** Fisher merging (Matena & Raffel, 2022) requires over 256 data points per task to estimate Fisher information. RegMean (Jin et al., 2022) requires more than 1600 data points per task to compute the inner product matrices effectively. AdaMerging (Yang et al., 2023) needs access to the full unlabeled test set for entropy minimization. Consensus TA/TIES (Wang et al., 2024b) requires a validation set to tune hyperparameters. In contrast, simple averaging (Wortsman et al., 2022a), task arithmetic (Ilharco et al., 2023) and TIES-Merging (Yadav et al., 2023) can be implemented without additional data. However, to achieve the best performance, both task arithmetic and TIES-Merging require tuning the hyperparameter $\alpha$ on a validation set.

## 3 Localize-and-Stitch

We now introduce our main algorithm. In Section 3.1, we start by outlining two insights that underpin effective model merging, accompanied by motivating examples. In Section 3.2 and Section 3.3, we provide a detailed description of two key components of the algorithm: localization and stitching.

### 3.1 Motivation and objectives

**Sparsity is important, but how to locate sparse regions is the key.** Previous research identifies that during the finetuning stage, a significant portion of parameter updates is redundant, introducing interference in model merging (Yadav et al., 2023). This underscores the need for locating sparse regions to reduce such interference. While the importance of sparsity is recognized, strategies for achieving it remain underexplored. Earlier approaches typically identify sparse regions through random selection (Yu et al., 2023) or selecting regions with the top-$k$% largest magnitudes in task vectors (Yadav et al., 2023). However, they often fall short in identifying the most effective sparse regions for model merging. In Figure 2, we evaluate the efficacy of different localization methods across twelve language tasks, comparing the quality of their localized regions (specified by the binary mask $\gamma_i$). The performance is assessed on individual grafted models for each task, denoted as $\theta_{\text{pre}} + \gamma_i \odot \tau_i$, where $\tau_i$ is the task vector of the $i$-th task and $\odot$ is the element-wise product.

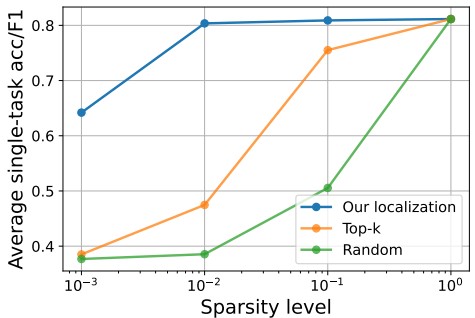 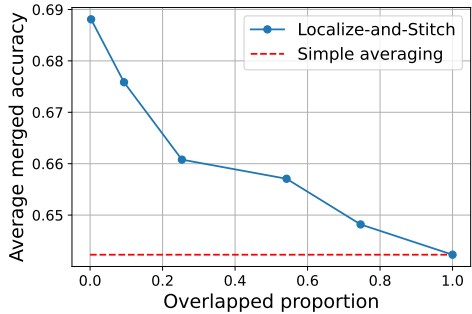

Figure 2: **Our method most effectively locates sparse regions essential for finetuned performance.** Sparsity level indicates the proportion of total parameters localized. By localizing only 1% of parameters (at sparsity level 0.01), our approach recovers 99% of the finetuned performance (at sparsity level 1).

Figure 3: **Merged models with more parameter overlap manifest more task conflicts, resulting in performance decrease.** The overlapped proportion is over the model's total parameter count. The simple averaging baseline is over all model parameters.

This grafted performance measures how well the finetuned skills are preserved when only keeping parameter updates during finetuning in localized regions. Unlike previous methods, we directly optimize the binary masks to maximally retain finetuned performance, detailed in Section 3.2. Our method significantly outperforms others, especially at lower sparsity levels. The strength of our approach lies in its precision in identifying small but informative regions, which is particularly advantageous for model merging.

**Sparse regions with less overlap reduce task conflicts.** Identifying the smallest possible regions with essential finetuned skills is key to minimizing potential conflicts among task vectors, as smaller localized regions naturally incur less overlap among tasks. With reduced overlap, each task can occupy its own, relatively disjoint localized region, thereby reducing task conflicts. This has the intuitive explanation that when two conflicting tasks share highly overlapping localized regions, they will compete to steer the parameters within these regions to their advantage in the merged model, leading to performance degradation. We demonstrate this by a case study (Figure 3) on merging models finetuned on two conflicting language tasks: QNLI (Wang et al., 2018) and MNLI (Bowman et al., 2015). QNLI involves predicting whether a context contains the answer to the given question, and MNLI involves predicting text entailment given a sentence pair. These tasks are conflicting, manifested by a noticeable performance decline for both tasks when using simple averaging to merge the corresponding finetuned models. However, if the localized regions are small yet sufficiently informative about their respective tasks, the reduced overlap between these regions decreases task conflicts and enhances overall performance after merging. In other words, as long as the localized region contains sufficient task-specific knowledge, including more parameters than necessary in them only introduces additional task interference.

### 3.2 Localization

Motivated by the importance of locating informative yet small sparse regions, we outline two objectives for localization in finetuned models: i) the regions should encapsulate essential skills acquired during finetuning, and ii) they should contain minimal number of parameters.

The objectives are grounded in the findings of Panigrahi et al. (2023), which demonstrates that skills developed during finetuning are localizable. Specifically, grafting a small subset of finetuned parameters onto the pretrained model can almost fully recover the performance of the finetuned model. Panigrahi et al. (2023) propose the following optimization problem to identify the localized parameters, and we adapt it in the model merging setting. With the constraint on the sparsity level $s$, on the $i$-th task with the loss function $\ell_i$ and task vector $\tau_i$, we optimize for a binary mask $\gamma_i$ such that only adding up the masked portion of the task vector onto the pretrained model performs well on the $i$-th task

$$\gamma_i = \underset{\gamma \in \{0,1\}^d : \|\gamma\|_0 \leq s}{\arg\min} \ell_i(\theta_{\text{pre}} + \gamma \odot \tau_i),$$

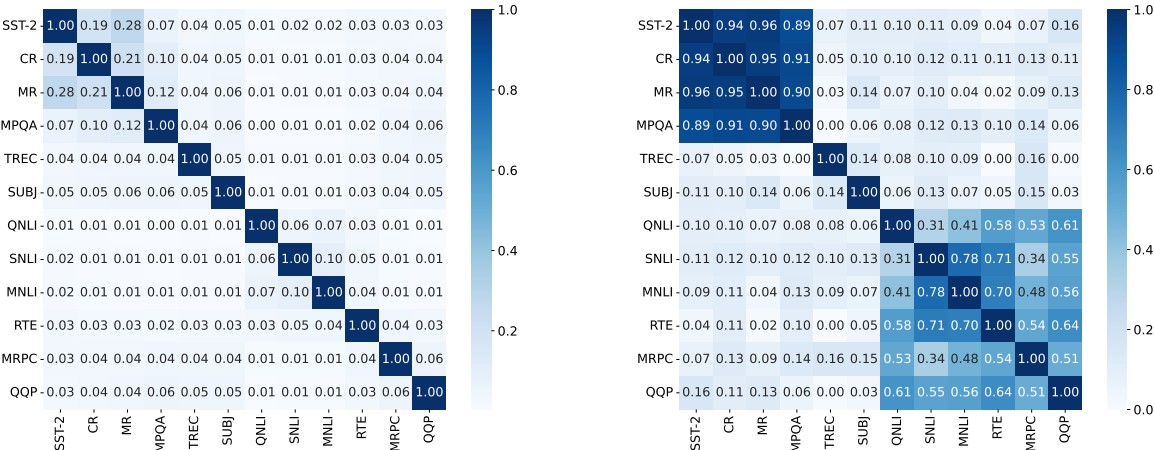

(a) Jaccard similarity of pairwise task masks.  (b) Cosine similarity of masked task vectors.

Figure 4: **Our localized regions (each task with $1\%$ of total parameters) have little pairwise overlap, with the majority of Jaccard similarity below $5\%$.** The sentiment classification tasks (SST-2, CR, MR, MPQA) have relatively large overlap because they share similar skills in the overlapping regions, and we verify this by showing that they have high cosine similarity of masked task vectors.

where $\odot$ denotes the element-wise product. For the ease of optimization, we follow Panigrahi et al. (2023) to reparametrize the binary mask $\gamma$ as a real-valued vector $S$. To control the sparsity, we additionally relax the $L_0$ sparsity constraint to be $L_1$. As a result, the optimization is reformulated as

$$S_i = \arg\min_{S \in \mathbb{R}^d} \ell_i \left( \theta_{\text{pre}} + \sigma(S) \odot \tau_i \right) + \lambda \|\sigma(S)\|_1, \tag{1}$$

where $\sigma$ is the sigmoid function, and $\lambda$ controls the strength of the $L_1$ regularization. At the end of the optimization, we round $\sigma(S_i)$ to be binary.

In comparison, Panigrahi et al. (2023) uses the following formulation

$$S_i = \arg\min_{S \in \mathbb{R}^d} \ell_i(\gamma \odot \theta_{\text{ft}} + (1 - \gamma) \odot \theta_{\text{pre}}),$$
$$\gamma := \gamma_{base} \odot (1 - \sigma(S)) + (1 - \gamma_{base}) \odot \sigma(S), \tag{2}$$

where $\gamma_{base}$ is the top-$k\%$ largest elements in the task vector, which serves as an initialization for the optimization. There are two main advantages of formulation 1 over 2. Firstly, our formulation of $S$ is more straightforward, as we directly have $\gamma = \sigma(S)$. In contrast, $S$ in Equation (2) serves as a selector of whether to take the value from $\gamma_{base}$, leading to more complex computation. Secondly, our approach uses the $L_1$ constraint to control the sparsity in a more fine-grained manner, while Equation (2) does not have this constraint, and they control the sparsity via early stopping instead. A detailed empirical performance comparison between our localization technique and the one in (Panigrahi et al., 2023) is presented in Appendix E.

Note that the optimization is highly efficient, requiring as few as 8-shot data with 10 epochs of training using SGD. A detailed ablation on the impact of data availability on the mask quality is shown in Section 4.4.

**Interpretation of task relationships.** In Figure 4, we validate that our localization method effectively identifies task-specific regions with minimal overlap. In Figure 4a, we report the Jaccard similarity of all pairwise task masks, namely for each pair of masks $\gamma_i$ and $\gamma_j$, we compute $|\gamma_i \cap \gamma_j|/|\gamma_i \cup \gamma_j|$. The majority of task pairs exhibit a Jaccard similarity below $5\%$, confirming minimal overlap. For the few pairs with Jaccard similarity larger than $10\%$ (upper left corner of Figure 4a), we further compute the cosine similarity of their masked task vectors in Figure 4b, and find that their cosine similarities are almost 1, indicating high agreement of the parameters within the overlapped regions. Since these four tasks are all sentiment

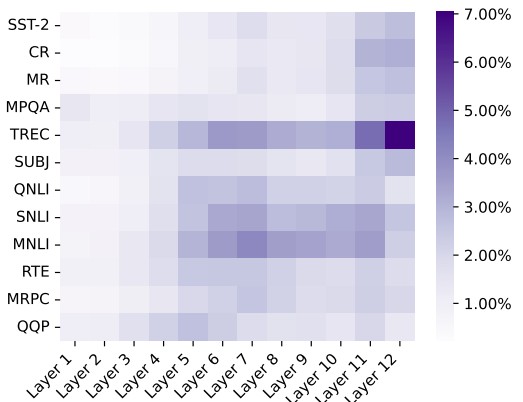
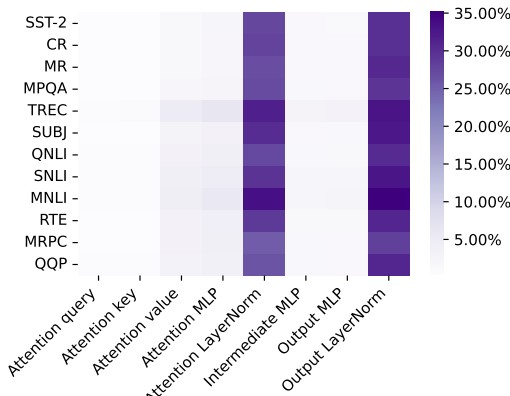

(a) Distribution of localized regions in different network layers in the RoBERTa-base model.

(b) Distribution of localized regions in different network components in the RoBERTa-base model.

Figure 5: **The localized regions are predominantly found in the LayerNorm parameters**, while different tasks are associated with different layers. The percentages represent the proportion of localized parameters in each component.

classification tasks, this phenomenon intuitively suggests a shared skill set across the tasks, located in the overlapped regions. We elaborate our resolution for the overlapped regions in Section 3.3

It is important to note that the meaningfulness of cosine similarity depends heavily on the presence of a substantial overlap, as indicated by Jaccard similarity. In cases where overlap is minimal, high cosine similarity might imply a strong relationship due to well-aligned parameters. Yet, this interpretation could be misleading without the broader context provided by Jaccard similarity, which could reveal that the actual interaction between the tasks is limited. This understanding is crucial for accurately assessing the nature of the relationships between tasks based on their localized parameters.

**Distribution of localized regions.** We analyze the distribution of the localized regions for language tasks in RoBERTa-base models Figure 5, both in terms of the layer index and the transformer components. For the layers, different tasks seem to occupy different layers, although the earlier layers in the network seldomly appear in the localized regions. Interestingly, most of the localized regions concentrate in the LayerNorm (Ba, 2016) parameters. This pattern can possibly be attributed to a distribution shift observed in the finetuning data compared to the pretraining data, necessitating adjustments to the LayerNorm parameters to accommodate this shift. The same plots for GPT2-XL and ViT can be found in Appendix B, and the findings hold true for those models as well.

**Localization without validation data.** In the rare cases where no labeled data is available, we adopt a similar strategy as the "Trim" step in TIES-Merging (Yadav et al., 2023), which selects positions in task vectors with the top-$k$% largest magnitudes, i.e., the parameters changed the most during finetuning. We refer to our approach as `Dataless Localize-and-Stitch`. As shown in Figure 2, to match the performance of localization with validation data, the dataless version typically requires locating larger regions. This expansion is necessary to encapsulate sufficient skills acquired during finetuning, but it also leads to increased task conflicts. Nevertheless, in Section 4, we show that our dataless version still outperforms all other methods that do not require additional validation data.

Despite the similarity, there are two key differences between `Dataless Localize-and-Stitch` and TIES: **i) Smaller localized region:** We find selecting the top-5% of parameters is sufficient for our pipeline, compared to the top-20% recommended by TIES-Merging. Our smaller selected region incurs less overlapping, leading to reduced task interference. Note that this is not the only advantage of our approach, as reducing the threshold in TIES to be 5% does not yield an improved performance as demonstrated in Appendix C (Tables 12 and 13). **ii) Better merging performance**: We use "Stitching" described in the next section for merging the localized regions, instead of the "Elect" procedure in TIES. The "Elect" approach in TIES

might work well when overlapping regions involve a larger number of tasks, allowing sign determination and selective averaging to better capture a consensus among task vectors. However, when only two tasks are involved in the overlapping regions (which is often the case as shown in Figure 12 in Appendix C), TIES may only retain parameters predominantly from the task with the larger magnitude at each position. In such scenarios, important parameters for both tasks could be alternately ignored, impairing the overall efficacy in maintaining crucial task-specific information, particularly in tightly contested regions. We provide a more detailed discussion about the advantages of `Dataless Localize-and-Stitch` over TIES-Merging with empirical evidence in Appendix C.

### 3.3 Stitching

After obtaining the binary mask for each task, we integrate these masks, and apply them to task vectors to construct the merged model. Given the sparsity, the masks generally activate different positions for different tasks, minimizing overlap. However, in instances where overlaps occur – that is, where multiple tasks share the same mask positions – we address this by averaging the parameters in these regions. Specifically, for each final mask $\gamma_i'$, the value at the $k$-th position, denoted $\gamma_i'[k]$, is calculated as the reciprocal of the total number of tasks that have a mask value of 1 at that position; if the original mask value $\gamma_i[k]$ is 0, it remains 0 in the processed mask: $\gamma_i'[k] = \gamma_i[k] \Big/ \left( \sum_{j=1}^{n} \gamma_j[k] \right)$. After we obtain the processed masks $\{\gamma_i'\}_{i=1}^{n}$, we apply them to the task vectors and stitch the masked task vectors to get the final merged model

$$\theta_{\mathrm{merged}} = \theta_{\mathrm{pre}} + \sum_{i}^{n} \left( \gamma_i' \odot \tau_i \right).$$

The complete algorithm is presented in Algorithm 1. Note that our stitching step **does not involve tuning the scaling factors** $\alpha$ as other methods mentioned in Section 2, which typically requires grid search or other optimization strategies for tuning. This distinction simplifies our method and avoids the computational overhead. A comparison of runtime is provided in Appendix B.

**Remark.** In `Localize-and-Stitch`, the majority of computational overhead lies in the localization step, while the subsequent stitching process is notably efficient. This distribution of workload is ideal because the more intensive localization step is performed separately on each individual finetuned model. This property provides simple extension in continual learning settings: When integrating a new model into the existing merged model (or updating any of the merged models), only the localization step for that new model incurs a cost, followed by stitching. This is in contrast to most model merging methods Jin et al. (2022); Ilharco et al. (2023); Yadav et al. (2023); Yang et al. (2023) which necessitate restarting the whole merging process, as the scaling factors of task vectors are tuned or learned based on the performance of the merged model. We provide an experiment of continual learning in Section 4.4 to illustrate this advantage.

---

**Algorithm 1** Localize-and-Stitch

**Input:** Pretrained model $\theta_{\mathrm{pre}}$, finetuned models $\{\theta_{\mathrm{ft}}^{(i)}\}_{i=1}^{n}$, regularization coefficient $\lambda$, magnitude threshold $k$

**Output:** Merged model $\theta_{\mathrm{merged}}$, binary masks $\{\gamma_i\}_{i=1}^{n}$

```
// Step 1: Localization
```
**for** $i = 1, 2, \cdots, n$ **do**
    Compute the task vector $\tau_i = \theta_{\mathrm{ft}}^{(i)} - \theta_{\mathrm{pre}}$
    **if** validation data available **then**
        $S_i = \arg\min_S \ell_i \left( \theta_{\mathrm{pre}} + \sigma(S) \odot \tau_i \right) + \lambda \| \sigma(S) \|_1$
        `// make the mask binary` $\gamma_i = \mathrm{round}(\sigma(S_i))$
    **else**
        `// Dataless Localization`
        $\gamma_i[|\tau_i| > \text{top-}k(|\tau_i|)] = 1$ otherwise 0
    **end if**
**end for**
```
// Step 2: Stitching
```
**for** $i = 1, 2, \cdots, n$ **do**
    **for** $k = 1, 2, \cdots, d$ **do**
        `// take average for overlaps`
        $\gamma_i'[k] = \gamma_i[k] / \left( \sum_{j=1}^{n} \gamma_j[k] \right)$
    **end for**
**end for**
**return** $\theta_{\mathrm{merged}} = \theta_{\mathrm{pre}} + \sum_{i=1}^{n} \left( \gamma_i' \odot \tau_i \right)$

---

Table 2: Multi-task performance of merged RoBERTa-base models on twelve NLP tasks and merged CLIP ViT-B/32 models on eight vision tasks. The reported performance metric is average absolute and normalized (in parentheses) accuracy, with the only exception of MRPC which is evaluated via F1 score due to data imbalance.

| Method | Validation data | Acc/F1 on 12 NLP tasks | Acc on 8 vision tasks |
|---|---|---|---|
| Single-task finetuned | - | 0.811 | 0.905 |
| Simple averaging (Wortsman et al., 2022a) | ✗ | $0.563_{(0.696)}$ | $0.658_{(0.731)}$ |
| Task artihmetic (Ilharco et al., 2023) | ✗ | $0.626_{(0.777)}$ | $0.692_{(0.758)}$ |
| TIES (Yadav et al., 2023) | ✗ | $0.600_{(0.743)}$ | $0.725_{(0.796)}$ |
| **Dataless Localize-and-Stitch** | ✗ | $\mathbf{0.734}_{(\mathbf{0.911})}$ | $\mathbf{0.740}_{(\mathbf{0.818})}$ |
| Task arithmetic (Ilharco et al., 2023) | ✓ | $0.675_{(0.840)}$ | $0.701_{(0.778)}$ |
| TIES (Yadav et al., 2023) | ✓ | $0.621_{(0.772)}$ | $0.736_{(0.812)}$ |
| Fisher merging (Matena & Raffel, 2022) | ✓ | $0.690_{(0.863)}$ | $0.683_{(0.763)}$ |
| RegMean (Jin et al., 2022) | ✓ | $0.739_{(0.911)}$ | $0.718_{(0.792)}$ |
| AdaMerging (Yang et al., 2023) | ✓ | $0.637_{(0.790)}$ | $\mathbf{0.801}_{(\mathbf{0.880})}$ |
| Concensus TA (Wang et al., 2024b) | ✓ | $0.715_{(0.889)}$ | $0.737_{(0.810)}$ |
| Consensus TIES (Wang et al., 2024b) | ✓ | $0.695_{(0.863)}$ | $0.748_{(0.822)}$ |
| **Localize-and-Stitch** | ✓ | $\mathbf{0.759}_{(\mathbf{0.936})}$ | $0.799_{(\mathbf{0.880})}$ |

## 4 Experiments

We evaluate `Localize-and-Stitch` with baselines described in Section 2 under various experimental settings. Our localization step is performed with 64-shot validation data, and the sparsity is chosen to be 1%. In the dataless version, the sparsity is chosen to be 5%.

### 4.1 Merging finetuned encoder-based language models

Following Panigrahi et al. (2023), we finetune the RoBERTa-base (Liu et al., 2019) model on twelve GLUE (Wang et al., 2018) tasks. Specifically, the dataset suite includes six single-sentence tasks (SST-2 (Socher et al., 2013), CR (Hu & Liu, 2004), MR (Pang & Lee, 2005), MPQA (Wiebe et al., 2005), TREC (Voorhees et al., 1999), SUBJ (Pang & Lee, 2004)) and six pairwise-sentence tasks (QNLI (Wang et al., 2018), SNLI (Bowman et al., 2015), MNLI (Williams et al., 2017), RTE (Wang et al., 2018), MRPC Dolan & Brockett (2005), QQP (Iyer et al.)). The dataset details can be found in Appendix G.

We present the results in Table 2, and leave the detailed per-task results in Table 4 of Appendix A. The table is structured into three blocks for clarity: the upper block displays the performance of individually finetuned models for each task, the middle block lists algorithms that operate without the need for validation data, whereas the lower block includes algorithms that require validation data. Note that both the middle and lower blocks contain Task arithmetic and TIES because they are applicable with or without data. Both algorithms are able to utilize validation data to tune the merging coefficients $\alpha$, as in $\theta_{\text{merged}} = \theta_{\text{pre}} + \alpha \sum_{i=1}^{n} \tau_i$. We follow common practice to search over $\{0.1, 0.2, \cdots, 1\}$ to obtain the optimal coefficients. When no validation data is available, we use their suggested merging coefficient of 0.4.

From Table 2, regardless of data availability, our approach consistently outperforms other baselines. Notably, the dataless version of our algorithm provides more than 10% performance increase over task arithmetic and surpasses methods that depend on validation data (Fisher merging and AdaMerging), demonstrating its effectiveness. Note that TIES-Merging, although sharing one similar step with our dataless version, performs worse than task arithmetic. This performance decrease is also observed in similar language evaluation settings with similar model size Yadav et al. (2023). This phenomenon can be attributed to the two possible factors we identify in Section 3.3: i) the larger localized regions of TIES potentially lead to more task conflicts; ii) the sign election mechanism it employs tends to be less effective in overlapping regions that involve only a few tasks, particularly when just two are present. This can lead to suboptimal retention of essential task-specific information. We provide further analysis comparing `Dataless Localize-and-Stitch` and TIES-Merging in Appendix C.

Table 3: Multi-task performance of merged GPT2-XL models (normalized metrics shown in the parentheses).

| Task | MMLU (5-shot Acc) | TruthfulQA (MC2) | ARC (Acc) | Average |
|---|---|---|---|---|
| Single-task | 0.273 | 0.488 | 0.472 | 0.411 |
| Simple averaging (Wortsman et al., 2022a) | 0.234 $_{(0.857)}$ | 0.390 $_{(0.799)}$ | 0.406 $_{(0.860)}$ | 0.344 $_{(0.839)}$ |
| Task arithmetic (Ilharco et al., 2023) | 0.234 $_{(0.857)}$ | 0.390 $_{(0.799)}$ | 0.399 $_{(0.845)}$ | 0.341 $_{(0.834)}$ |
| TIES (Yadav et al., 2023) | 0.233 $_{(0.853)}$ | 0.448 $_{(0.918)}$ | 0.310 $_{(0.657)}$ | 0.330 $_{(0.809)}$ |
| Consensus TA (Wang et al., 2024b) | 0.234 $_{(0.857)}$ | 0.412 $_{(0.844)}$ | 0.373 $_{(0.790)}$ | 0.340 $_{(0.831)}$ |
| Consensus TIES (Wang et al., 2024b) | 0.232 $_{(0.850)}$ | 0.395 $_{(0.809)}$ | 0.386 $_{(0.818)}$ | 0.337 $_{(0.826)}$ |
| **Dataless Localize-and-Stitch** | 0.256 $_{(0.938)}$ | 0.394 $_{(0.807)}$ | 0.427 $_{(0.905)}$ | 0.359 $_{(0.883)}$ |
| **Localize-and-Stitch** | 0.247 $_{(0.905)}$ | 0.388 $_{(0.795)}$ | 0.467 $_{(0.989)}$ | **0.367** $_{(0.896)}$ |

## 4.2 Merging finetuned vision models

Following the practice in Ilharco et al. (2023), we finetune the CLIP (Radford et al., 2021) image encoder with the ViT-B/32 (Dosovitskiy et al., 2021) architecture on eight image classification tasks, incorporating diverse categories of images such as remote sensing, satellite images and traffic signs. Specifically, the dataset suite includes SUN397 (Xiao et al., 2016), Stanford Cars Krause et al. (2013), RESISC45 (Cheng et al., 2017), EuroSAT (Helber et al., 2019), SVHN (Netzer et al., 2011), GTRSB (Stallkamp et al., 2011), MNIST (LeCun et al., 2010) and DTD (Cimpoi et al., 2014). The details of each dataset can be found in Appendix G.

We present the results in Table 2, and leave the detailed per-task results in Table 5 of Appendix A. Similarly, even in the absence of validation data, the dataless version of our approach can outperform methods requiring validation data (Fisher merging and RegMean). When validation data is available, our method also demonstrates competitive performance. Note that AdaMerging, while achieving similar results as ours, imposes more demanding data availability requirement, and incurs higher computational cost. It necessitates entropy minimization across the entire (unlabeled) test set, rendering it approximately 15 times slower than our method.

## 4.3 Merging finetuned decoder-based language models

Compared with encoder-only language models, decoder-based language models benefit from increased number of parameters and perform well on complicated generative tasks. We use GPT2-XL (Radford et al., 2019) as the base model, and obtain three supervised finetuned checkpoints from the Hugging Face model hub (Wolf et al., 2020), each tuned for distinct functionalities: general reasoning, scientific knowledge and truthfulness respectively. Further details about these models are specified in Appendix F.

To assess these models, we use MMLU (Hendrycks et al., 2021), ARC (Clark et al., 2018) and TruthfulQA (Lin et al., 2021) as evaluation datasets for the respective domains. Unlike datasets in the previous section, these are typically used in their entirety for evaluation, without a designated train-test split. However, using these datasets for both evaluation and localization could lead to data leakage. To prevent this, we use data from three surrogate datasets with similar purposes for localization, namely Alpaca (Taori et al., 2023), GSM8K (Cobbe et al., 2021) and HotpotQA (Yang et al., 2018).

We compare our approach with other methods directly applicable in this setting in Table 3. Both versions of `Localize-and-Stitch` noticeably outperforms other baselines. The result verifies that even for complex generative tasks, skills can still be localized within tiny regions of finetuned models. Moreover, this shows that good localization performance can be achieved without access to the original finetuning data; using similar data from other sources also suffices. This aligns with the finding from (Panigrahi et al., 2023) that localized regions exhibit transfer among similar tasks, meaning that a localized region for one task can facilitate performance in related tasks. This further reduces the dependency on data availability, making our approach more versatile. Overall, these findings highlight the capability of `Localize-and-Stitch` to integrate the strengths from multiple language models, demonstrating its effectiveness across a variety of linguistic challenges.

## 4.4 Empirical analysis

**Sparsity-performance trade-off.** In the localization step, we optimize for two competing objectives: identifying regions containing sufficient finetuned skills, and minimizing the number of parameters involved.

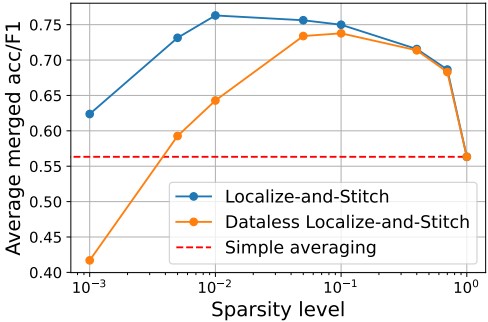
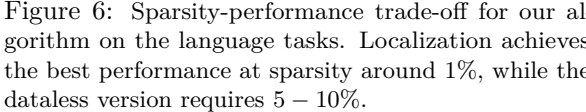

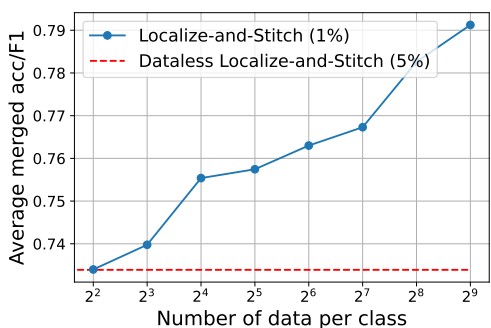

Figure 6: Sparsity-performance trade-off for our algorithm on the language tasks. Localization achieves the best performance at sparsity around 1%, while the dataless version requires $5 - 10\%$.

Figure 7: **With only 8-shot data, the performance of our algorithm improves over the dataless version.** With more data available, the performance of our method continues to increase. Numbers in brackets represent sparsity levels for each method.

Here, we study the trade-off by presenting the performance of our approach on the language tasks at different sparsity levels in Figure 6. Across all models and tasks tested, we observe that a sparsity level around 1% yields the best results using our localization method, whereas dataless localization requires $5 - 10\%$. When the localized regions are too small to retain adequate finetuned knowledge, the benefit of less overlap is diminished. Conversely, when the localized regions are too large, although the regions possess sufficient finetuned knowledge, the increased overlap among task-specific regions leads to more task interference.

**Effect of data availability.** We present the performance of our method across various data availability scenarios with a localization region of 1% (Figure 7) on the language tasks. One clear trend is that with more data, the quality of localization improves, resulting in enhanced performance of the merged model. Notably, even with as few as 8-shot data, the merged performance surpasses that of the dataless approach, highlighting its effectiveness under constrained data conditions.

**Avoid forgetting of pretrained knowledge.** Pretrained models contain rich and generalizable information, derived from their diverse training data. However, finetuning often incurs catastrophic forgetting of skills in the pretrained model (He et al., 2021; Luo et al., 2023), which is carried over when these finetuned models are merged. As our method makes minimal change to the pretrained model, such forgetting is substantially mitigated. For instance, in the vision setting where we localize 1% parameters for each task, the total parameters changed in our merged model compared with the pretrained model is only around 7% for the 8 tasks as a result of minimal overlap. We evaluate the retention of pretrained capabilities with a general vision task that the pretrained CLIP model excels, namely zero-shot ImageNet classification (Deng et al., 2009), and report the results in Figure 8. Our method most effectively preserves the pretrained performance, while achieves superior performance on the eight merged tasks.

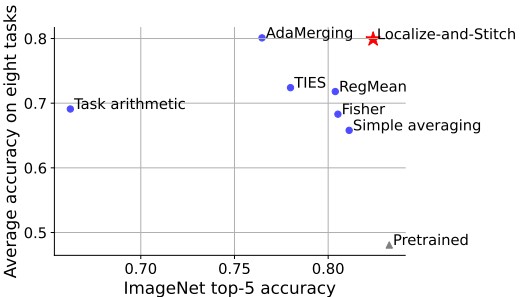

Figure 8: **Our method retains the pretrained skill the best** due to the minimal updates (7% of total parameters) to the pretrained model, while performs well on the eight merged tasks (upper right better).

**Model compression through localization.** Our localization approach enables a compact representation of the finetuned model. In our experiments, we find that localizing only 1% of the total parameters recovers over 99% of the performance achieved by single-task finetuning (full evaluation reported in Appendix B). The efficiency allows us to store only the masked task vectors for each task ($\gamma_i \odot \tau_i$) instead of the entire finetuned models, without a noticeable loss in performance. Given the sparsity of these masked task vectors,

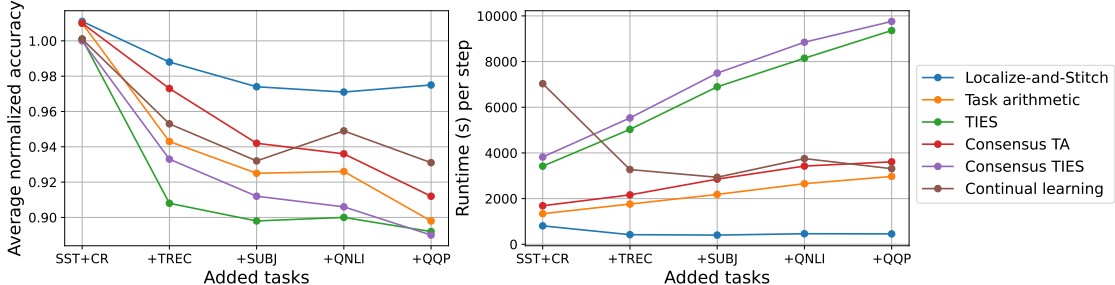

Figure 9: Comparison of performance (left) and efficiency (right) for continual learning. `Localize-and-Stitch` consistently outperforms baselines with generally constant runtime for each additional task.

we can store them in Compressed Sparse Row (CSR) format (Pissanetzky, 1984; Golub & Van Loan, 2013), which drastically reduces the model size to about 1% of the original. For example, a RoBERTa-base model, which typically requires $\sim 650$MB of memory to store, can be represented using only a $\sim 7$MB sparse matrix, achieving a memory reduction of 99%. Although we still need to store the full pretrained model, this storage overhead will be amortized with more finetuned models. This model compression, combined with the ease of update mentioned in Section 3.3, enables flexible composition of skills from multiple finetuned models with minimal storage and computational overhead.

**Continual learning.** As mentioned in Section 3.3, our approach is particularly efficient in the continual learning setting. To illustrate the efficiency, we start from merging the SST-2 and CR RoBERTa models, and incrementally add 4 more tasks to simulate a continual learning setting. The tasks are selected by representativeness (including sentiment analysis, sentence classification, NLI, etc). We compare our method against five baselines: task arithmetic, TIES, Consensus TA, Consensus TIES and continual training. Following common practice, the merging coefficients are tuned across $\{0.0, 0.1, ...0.9, 1.0\}$. The continual training results are obtained by sequential training on each task, using the model from the previous task as the starting point. The results from Figure 9 demonstrate that: i) **Performance**: `Localize-and-Stitch` consistently outperforms the baselines, with the performance margin increasing as more tasks are involved, showing its ability to reduce task interference. ii) **Runtime**: The runtime for other baselines increases with each added task due to the need for hyperparameter search from scratch and performance validations, while the runtime of `Localize-and-Stitch` generally remains constant, reflecting its efficiency in continual learning scenarios. While continual training performs better than other model merging baselines, it still falls short of `Localize-and-Stitch`. This suggests a potential avenue for enhancing continual learning: instead of finetuning on top of the last task model, it may be advantageous to first finetune the pretrained model on the new task and then merge it with the existing multi-task model on the old tasks. We leave further exploration of this approach for future work.

## 5 Related works

**Model merging.** Model merging aims at efficiently integrating multiple finetuned models into a single model that retains the capabilities of each. This approach enhances the efficiency, generalization and multi-task capabilities of finetuned models. In scenarios where models are *trained on the same task* with different training configurations, Singh & Jaggi (2020); Ainsworth et al. (2022); Jolicoeur-Martineau et al. (2024) show that merged models perform comparably to ensemble models but with significantly lower deployment costs. Additionally, Wortsman et al. (2022a;b) demonstrate that the merged model improves the out-of-distribution (OOD) robustness. When merging finetuned models *from different tasks*, the merged model can provide better initialization for new tasks (Choshen et al., 2022; Gueta et al., 2023). Finetuned models with different specialized skills can also be combined to enhance multi-task capabilities (Ilharco et al., 2023; Tam et al., 2023; Matena & Raffel, 2022; Jin et al., 2022; Yang et al., 2023; Yu et al., 2023; Wang et al., 2024b;a). More recently, a new line of work has emerged that uses a mixture of experts (MoE) strategy (Jiang et al., 2023; Tang et al., 2024). Instead of a single unified model, the MoE approach incorporates routing mechanisms to direct inputs to task-specific networks. In this work, we primarily focus on merging specialized models into a single unified model for enhancing multi-task performance. Similar to the gradient conflict problem (Yu

et al., 2020; Liu et al., 2021) in multi-task learning, finetuned models also manifest conflict when merged together, and our method provides an effective solution to this problem.

Our approach stands out with four key advantages: i) Localized merging: Instead of global merging, we localize merging to specific regions with finetuned skills, effectively decreasing task conflicts. ii) Simplified process: Existing works often require computationally intensive grid search or optimization to determine the optimal merging coefficients, while our stitching procedure does not have the requirement. iii) Data flexibility: Our method works with or without validation data, and provides competitive results in various data availability scenarios. iv) Benefits beyond model merging: This includes interpretability of task relations, model compression and preservation of pretrained knowledge.

**Knowledge attribution.** Recent works find that knowledge contained in language models is localizable, meaning that model behavior can be attributed to small subsets of model parameters. One line of work identifies such regions to edit the knowledge contained in the networks. Sundararajan et al. (2017) uses integrated gradients for knowledge attribution, which measures how sensitive each neuron's gradient is to the change of input. Dai et al. (2021) applies integrated gradients to edit factual knowledge contained in BERT models. However, the relationship between the editing success and the localized regions remains unclear (Hase et al., 2024). Knowledge attribution can also be applied for enhancing interpretability. Vig et al. (2020) applies causal mediation analysis (Pearl, 2022) to identify individual neurons contributing to gender bias.

Recently, Panigrahi et al. (2023) optimizes for a binary mask to localize skills contained in finetuned language models. There are two key differences between our localization formulation and theirs. Firstly, our formulation of $S$ is more straightforward, as we directly have $\gamma = \sigma(S)$ in Equation (1). In contrast, Panigrahi et al. (2023) uses $S$ as a selector of whether to take the value from the initial mask, leading to more complex computation. Secondly, our approach uses the $L_1$ constraint to control the sparsity in a more fine-grained manner, while Panigrahi et al. (2023) does not have this constraint, and controls sparsity via early stopping. We empirically show that our localization formulation identifies regions with improved quality in Appendix E.

**Pruning.** Similar to localization, pruning is a strategy to identify key regions in the parameter space that are important to model performance. Han et al. (2015a;b) propose magnitude pruning, which preserves weights with high magnitudes, and the method inspires various variants (Zhu & Gupta, 2017; Paganini & Forde, 2020; Zafrir et al., 2021). Parameter significance measured by performance sensitivity is another effective criteria for identifying important parameters (Sanh et al., 2020; Liang et al., 2021; Zhang et al., 2022). For instance, Lee et al. (2018) proposes SNIP score, which computes the change of loss when each neuron is set to 0. Pruning methods are widely applied to modern large language models (Zhang et al., 2023; Sun et al., 2023; Xia et al., 2023; Zhao et al., 2024; Cheng et al., 2024).

The primary distinction between pruning and localization lies in their treatment of parameters outside the identified regions. In pruning, these parameters are set to zero, effectively removing them, allowing the pruned network to function as a standalone model. Conversely, in localization, parameters outside the localized regions are retained at their pretrained values, requiring the localized regions to be combined with the pretrained model for deployment. Despite these differences, the conceptual overlap between them suggests that pruning techniques could be adapted for localization. Exploring such adaptations presents an interesting direction for future work.

## 6  Conclusion

In this work, we study the problem of task interference in the context of model merging. We find that globally merging models typically leads to task interference, due to the parameter redundancy in task vectors. To tackle this challenge, we introduce `Localize-and-Stitch`, which performs localized merging via sparse task arithmetic. We first identify tiny regions in the finetuned models that contain essential skills acquired during finetuning, and stitch only those regions back onto the pretrained model. Empirical evaluation on various vision and language benchmarks validate the effectiveness of our approach. Beyond model merging, our approach performs effective model compression, which compresses the model size to be 1% of the original without sacrificing performance. Additionally, `Localize-and-Stitch` also excels at retaining the pretrained knowledge. Overall, our approach offers a novel pathway for flexible and continual skills composition from finetuned models with minimal storage and computational overhead.

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

# A    Full experimental results

Table 4: Multi-task accuracy ([†]F1) of merged RoBERTa-base models on twelve NLP tasks.

| Task | SST-2 | CR | MR | MPQA | TREC | SUBJ | QNLI | SNLI | MNLI | RTE | MRPC[†] | QQP | Average Acc/F1 |
|---|---|---|---|---|---|---|---|---|---|---|---|---|---|
| Single-task finetuned | 0.898 | 0.894 | 0.844 | 0.848 | 0.938 | 0.931 | 0.764 | 0.791 | 0.706 | 0.643 | 0.766 | 0.716 | 0.811 |
| Simple averaging (Wortsman et al., 2022a) | 0.857 | 0.851 | 0.829 | 0.688 | 0.304 | 0.478 | 0.508 | 0.452 | 0.452 | 0.563 | 0.311 | 0.469 | 0.563 |
| Task arithmetic (Ilharco et al., 2023) | 0.846 | 0.856 | 0.769 | 0.810 | 0.156 | 0.584 | 0.607 | 0.538 | 0.403 | 0.539 | 0.822 | 0.581 | 0.626 |
| TIES (Yadav et al., 2023) | 0.805 | 0.805 | 0.728 | 0.791 | 0.226 | 0.549 | 0.552 | 0.501 | 0.379 | 0.477 | 0.816 | 0.572 | 0.600 |
| **Dataless Localize-and-Stitch** | 0.909 | 0.907 | 0.864 | 0.821 | 0.462 | 0.762 | 0.558 | 0.690 | 0.618 | 0.688 | 0.837 | 0.693 | **0.734** |
| Task arithmetic (Ilharco et al., 2023) | 0.885 | 0.882 | 0.803 | 0.829 | 0.320 | 0.610 | 0.620 | 0.561 | 0.495 | 0.656 | 0.828 | 0.623 | 0.675 |
| TIES (Yadav et al., 2023) | 0.886 | 0.88 | 0.852 | 0.835 | 0.226 | 0.482 | 0.548 | 0.359 | 0.397 | 0.594 | 0.794 | 0.603 | 0.621 |
| Fisher merging (Matena & Raffel, 2022) | 0.900 | 0.898 | 0.837 | 0.758 | 0.260 | 0.546 | 0.542 | 0.725 | 0.652 | 0.656 | 0.833 | 0.677 | 0.690 |
| RegMean (Jin et al., 2022) | 0.897 | 0.897 | 0.847 | 0.826 | 0.730 | 0.791 | 0.559 | 0.683 | 0.568 | 0.638 | 0.794 | 0.642 | 0.739 |
| AdaMerging (Yang et al., 2023) | 0.850 | 0.861 | 0.778 | 0.815 | 0.230 | 0.595 | 0.612 | 0.541 | 0.404 | 0.547 | 0.822 | 0.588 | 0.637 |
| Consensus TA (Wang et al., 2024b) | 0.892 | 0.894 | 0.866 | 0.882 | 0.376 | 0.596 | 0.714 | 0.665 | 0.520 | 0.630 | 0.881 | 0.664 | 0.715 |
| Consensus TIES (Wang et al., 2024b) | 0.898 | 0.893 | 0.847 | 0.862 | 0.314 | 0.631 | 0.689 | 0.604 | 0.461 | 0.630 | 0.862 | 0.644 | 0.695 |
| **Localize-and-Stitch** | 0.896 | 0.896 | 0.849 | 0.828 | 0.782 | 0.820 | 0.734 | 0.621 | 0.580 | 0.633 | 0.820 | 0.651 | **0.759** |

Table 5: Multi-task accuracy of merged CLIP ViT-B/32 models on eight vision tasks.

| Task | SUN397 | Cars | RESISC45 | EuroSAT | SVHN | GTSRB | MNIST | DTD | Average Acc |
|---|---|---|---|---|---|---|---|---|---|
| Single-task finetuned | 0.753 | 0.777 | 0.961 | 0.997 | 0.975 | 0.987 | 0.997 | 0.794 | 0.905 |
| Simple averaging (Wortsman et al., 2022a) | 0.653 | 0.634 | 0.714 | 0.717 | 0.642 | 0.528 | 0.875 | 0.501 | 0.658 |
| Task arithmetic (Ilharco et al., 2023) | 0.552 | 0.549 | 0.667 | 0.789 | 0.802 | 0.697 | 0.973 | 0.504 | 0.692 |
| TIES (Yadav et al., 2023) | 0.598 | 0.586 | 0.707 | 0.797 | 0.862 | 0.721 | 0.983 | 0.542 | 0.725 |
| **Dataless Localize-and-Stitch** | 0.669 | 0.647 | 0.768 | 0.746 | 0.817 | 0.726 | 0.973 | 0.576 | **0.740** |
| Task arithmetic (Ilharco et al., 2023) | 0.638 | 0.621 | 0.720 | 0.776 | 0.744 | 0.651 | 0.970 | 0.522 | 0.701 |
| TIES (Yadav et al., 2023) | 0.648 | 0.629 | 0.743 | 0.789 | 0.831 | 0.714 | 0.976 | 0.562 | 0.736 |
| Fisher merging (Matena & Raffel, 2022) | 0.686 | 0.692 | 0.707 | 0.664 | 0.729 | 0.511 | 0.879 | 0.599 | 0.683 |
| RegMean (Jin et al., 2022) | 0.653 | 0.635 | 0.756 | 0.786 | 0.781 | 0.674 | 0.937 | 0.520 | 0.718 |
| AdaMerging (Yang et al., 2023) | 0.645 | 0.681 | 0.792 | 0.938 | 0.870 | 0.919 | 0.975 | 0.591 | **0.801** |
| Consensus TA (Wang et al., 2024b) | 0.639 | 0.641 | 0.755 | 0.794 | 0.816 | 0.699 | 0.980 | 0.551 | 0.737 |
| Consensus TIES (Wang et al., 2024b) | 0.623 | 0.622 | 0.745 | 0.800 | 0.877 | 0.775 | 0.986 | 0.553 | 0.748 |
| **Localize-and-Stitch** | 0.672 | 0.683 | 0.818 | 0.894 | 0.879 | 0.866 | 0.948 | 0.629 | **0.799** |

We present the per-task performance in Table 4 and Table 5. The reported normalized accuracy is computed by

$$\text{Normalized accuracy} = \frac{1}{T} \sum_{t \in [T]} \frac{\text{acc}_{\text{merged}}}{\text{acc}_{\text{finetuned}}}.$$

# B    More experiments

**More on task interference.** To assess our method's effectiveness on tasks with different task similarities, we created two subsets: i) Conceptually similar subset: Composed entirely of sentiment classification tasks

(SST-2, CR, MR, MPQA). ii) Conceptually dissimilar subset: Including tasks from different categories (SST-2 for sentiment classification, TREC for question classification, SUBJ for subjectivity, and MNLI for entailment). In Table 6, we report the average performance for each subset. In the similar subset, where tasks share similar skills, all merging methods perform equally well. However, in the dissimilar subset, where task skills differ and may even conflict, `Localize-and-Stitch` shows a significant advantage, demonstrating its ability to effectively resolve task interference.

Table 6: Performance comparison for merging similar and dissimilar tasks.

| Method | Average acc on similar subset | Average acc on dissimilar subset |
|---|---|---|
| Task arithmetic | 0.878 | 0.802 |
| TIES | 0.875 | 0.812 |
| Localize-and-Stitch | 0.880 | 0.831 |

**Where are the localized regions?** We analyze the distribution of the localized regions for both language and vision tasks in Figure 10, both in terms of the layer index and the transformer components. For the layers, different tasks seem to occupy different layers, although the earlier layers in the network seldomly appear in the localized regions. Interestingly, most of the localized regions concentrate in the LayerNorm parameters. This pattern can possibly be attributed to a distribution shift observed in the finetuning data compared to the pretraining data, necessitating adjustments to the LayerNorm parameters to accommodate this shift.

Table 7: Single-task grafted accuracy ([†]F1) of RoBERTa-base models on twelve NLP tasks.

| Task | SST-2 | CR | MR | MPQA | TREC | SUBJ | QNLI | SNLI | MNLI | RTE | MRPC[†] | QQP | Average acc/F1 |
|---|---|---|---|---|---|---|---|---|---|---|---|---|---|
| Single-task finetuned | 0.898 | 0.894 | 0.844 | 0.848 | 0.938 | 0.931 | 0.764 | 0.791 | 0.706 | 0.643 | 0.766 | 0.716 | 0.811 |
| Single-task grafted | 0.897 | 0.883 | 0.855 | 0.844 | 0.918 | 0.933 | 0.751 | 0.772 | 0.703 | 0.639 | 0.745 | 0.708 | 0.804 |
| Recovered proportion | 0.999 | 0.988 | 1.013 | 0.995 | 0.979 | 1.002 | 0.983 | 0.976 | 0.996 | 0.994 | 0.973 | 0.989 | 0.991 |

Table 8: Single-task grafted accuracy of GPT2-XL models on three NLP tasks.

| Task | MMLU (5-shot Acc) | TruthfulQA (MC2) | ARC (Acc) | Average |
|---|---|---|---|---|
| Single-task finetuned | 0.273 | 0.488 | 0.472 | 0.411 |
| Single-task grafted | 0.264 | 0.436 | 0.475 | 0.392 |
| Recovered proportion | 0.969 | 0.893 | 1.007 | 0.953 |

Table 9: Single-task grafted accuracy of CLIP ViT-B/32 models on eight vision tasks.

| Task | SUN397 | Cars | RESISC45 | EuroSAT | SVHN | GTSRB | MNIST | DTD | Average acc |
|---|---|---|---|---|---|---|---|---|---|
| Single-task finetuned | 0.753 | 0.777 | 0.961 | 0.997 | 0.975 | 0.987 | 0.997 | 0.794 | 0.905 |
| Single-task grafted | 0.731 | 0.772 | 0.955 | 0.989 | 0.963 | 0.973 | 0.996 | 0.781 | 0.895 |
| Recovered proportion | 0.970 | 0.993 | 0.994 | 0.992 | 0.988 | 0.985 | 0.999 | 0.983 | 0.989 |

**Full grafted performance.** We evaluate the quality of the localized regions by the grafted performance. For the $i$-th task, we only add up the localized regions in the task vectors back to the pretrained model, i.e., $\theta_{\mathrm{pre}} + \gamma_i \odot \tau_i$. The results with a localization region of 1% is shown in Tables 7 to 9. For almost all tasks, using only the tiny localized region recovers nearly 99% of the finetuned performance. For GPT2-XL, the performance is slightly worse because we cannot use the evaluation data for the localization step. However, the results are still strong even with surrogate datasets with similar purposes, demonstrating the flexibility and robustness of our algorithm. Overall, this shows that our localization approach is effective in locating regions containing essential skills acquired during finetuning, and the localized regions can be viewed as compact representations of the finetuned models.

**Effect of data availability.** Similar to Figure 7, we plot the same trend of our method across various data availability scenarios with a localization regions of 1% on the vision tasks as well (Figure 11). We can still

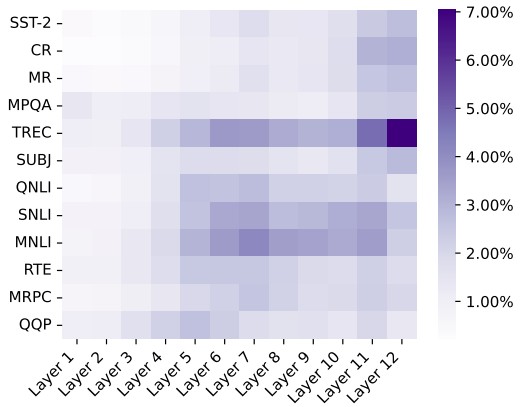

(a) Distribution of localized regions in different network layers for language tasks in the RoBERTa-base model.

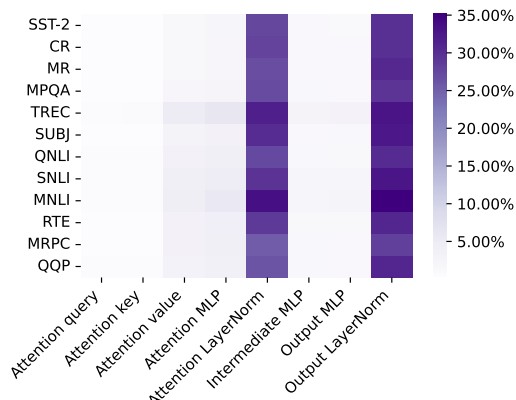

(b) Distribution of localized regions in different network components for language tasks in the RoBERTa-base model.

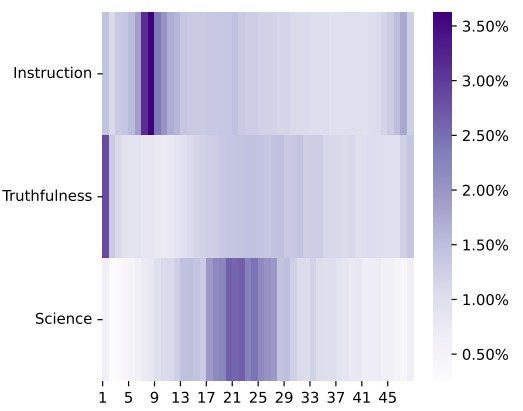

(c) Distribution of localized regions in different network layers for language tasks in the GPT2-XL model.

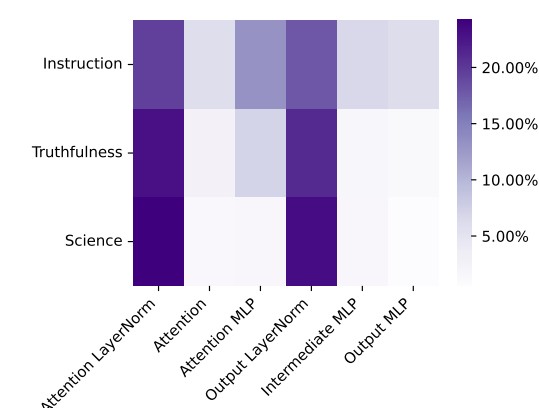

(d) Distribution of localized regions in different network components for language tasks in the GPT2-XL model.

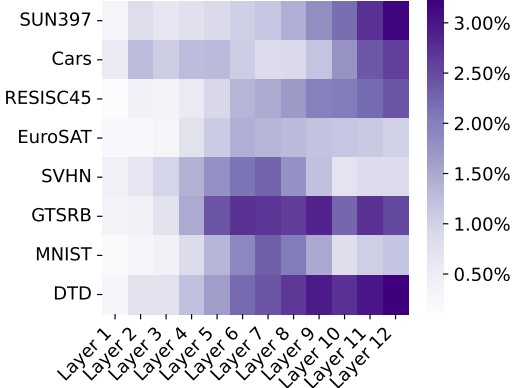

(e) Distribution of localized regions in different network layers for vision tasks.

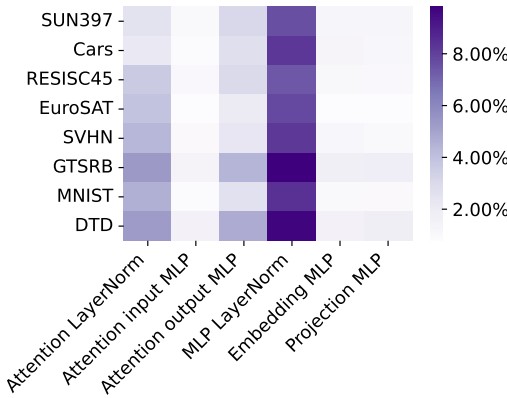

(f) Distribution of localized regions in different network components for vision tasks.

Figure 10: The localized regions are predominantly found in the LayerNorm parameters, while different tasks are associated with different layers. The percentages represent the proportion of localized parameters in each component.

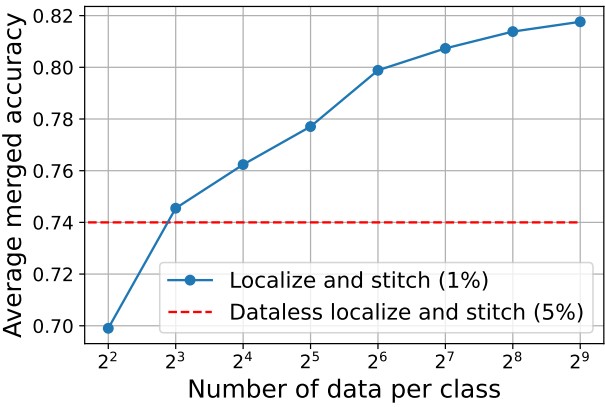

Figure 11: Merged performance versus available data in vision tasks.

see the clear pattern that the performance monotonically increases with more data available, and does not show saturation even with 512-shot data. In addition, with 8-shot data, the performance of localization improves over the dataless version, the same observation as in language tasks.

Table 11: Runtime for algorithms requiring data.

Table 10: Runtime for dataless algorithms.

| Method | Runtime (s) |
|---|---|
| Simple averaging | 189.17 |
| Task arithmetic | 186.32 |
| TIES-Merging | 350.47 |
| **Dataless Localize-and-Stitch** | 304.85 |

| Method | Runtime (s) |
|---|---|
| Fisher Merging | 293.33 |
| Task arithmetic (tuned) | 6562.14 |
| TIES-Merging (tuned) | 24042.43 |
| RegMean | 22987.54 |
| AdaMerging | 81326.57 |
| Consensus TA | 7361.52 |
| Consensus TIES | 26042.43 |
| **Localize-and-Stitch** | 5130.05 |

**Runtime.** We compare the runtime of `Localize-and-Stitch` and other baselines. We divide the algorithms into two categories: dataless and requiring data. Note that task arithmetic and TIES-Merging can fall in both categories, with the difference of whether performing hyperparameter tuning (scaling factor $\alpha$ for both, and sparsity for TIES). For the hyperparameter tuning, we follow the common practice in Ilharco et al. (2023); Yadav et al. (2023) to grid search over $\{0.1, 0.2 \cdots, 1\}$ for the scaling factor and $\{0.1, 0.2, 0.3\}$ for the sparsity.

We report the runtime in Tables 10 and 11 for merging twelve NLP tasks with RoBERTa-base. For the dataless algorithms, simple averaging and task arithmetic are very efficient, as they only involve arithmetic operations on the weights. Both TIES and our dataless version requires sorting the task vectors to get the top-$k\%$ largest elements, so the runtime is slower. Compared with TIES, we do not have the step for resolving sign conflicts, so it takes less time. For algorithms requiring data, Fisher merging is the most efficient, as it uses a diagonal estimate of the Fisher information matrices with little data (256 per task). Both task arithmetic and TIES-Merging show substantial time increase, as they need to do grid search on 9 and 27 hyperparameters respectively as well as evaluating on the validation data in each run. AdaMerging takes significantly more runtime to execute compared with others, and the reason could be that entropy minimization converges slowly, as we observe that AdaMerging requires around 500 epochs to converge. Compared with other algorithms, `Localize-and-Stitch` executes in a relatively short amount of time, showing its effectiveness.

## C  Comparison between `Dataless Localize-and-Stitch` and **TIES**-Merging

Due to the similarity of the dataless version of our approach with TIES-Merging (Yadav et al., 2023), we compare them in detail.

Table 12: Accuracy ([†]F1) comparison of `Dataless Localize-and-Stitch` and TIES on language tasks.

| Task | SST-2 | CR | MR | MPQA | TREC | SUBJ | QNLI | SNLI | MNLI | RTE | MRPC[†] | QQP | Average acc/F1 |
|---|---|---|---|---|---|---|---|---|---|---|---|---|---|
| **Dataless Localize-Stitch (5%)** | 0.909 | 0.907 | 0.864 | 0.821 | 0.462 | 0.762 | 0.558 | 0.690 | 0.618 | 0.688 | 0.837 | 0.693 | **0.734** |
| TIES (5%) | 0.858 | 0.837 | 0.822 | 0.712 | 0.142 | 0.290 | 0.467 | 0.191 | 0.271 | 0.438 | 0.743 | 0.358 | 0.510 |
| TIES (20%) | 0.805 | 0.805 | 0.728 | 0.791 | 0.226 | 0.549 | 0.552 | 0.501 | 0.379 | 0.477 | 0.816 | 0.572 | 0.600 |

Table 13: Accuracy comparison of `Dataless Localize-and-Stitch` and TIES on vision tasks.

| Task | SUN397 | Cars | RESISC45 | EuroSAT | SVHN | GTSRB | MNIST | DTD | Average acc |
|---|---|---|---|---|---|---|---|---|---|
| **Dataless Localize-Stitch (5%)** | 0.669 | 0.647 | 0.768 | 0.746 | 0.817 | 0.726 | 0.973 | 0.576 | **0.740** |
| TIES (5%) | 0.520 | 0.552 | 0.669 | 0.683 | 0.874 | 0.606 | 0.982 | 0.480 | 0.671 |
| TIES (20%) | 0.598 | 0.586 | 0.707 | 0.797 | 0.862 | 0.721 | 0.983 | 0.542 | 0.725 |

The first step of both algorithms is similar: select the top-$k$% largest positions in the task vector. The primary difference lies in the selection threshold: `Dataless Localize-and-Stitch` selects the top-5%, while TIES selects the top-20%. However, as shown in Section 3.1, we find that when a localized region already contains sufficient task-specific knowledge, including more parameters only introduces more task interference. This observation could partially explain our superior performance. However, this is not the only limitation in TIES, as reducing the threshold in TIES to be 5% does not yield an improved performance as demonstrated in Tables 12 and 13.

In the subsequent merging step, `Dataless Localize-and-Stitch` can be viewed as a simplified version of TIES. When dealing with overlap for the localized regions, `Dataless Localize-and-Stitch` simply averages the parameters in these overlapping area. On the other hand, TIES first sums positive and negative parameters separately at each overlapping position, and determines the dominant sign based on their total magnitudes, a process akin to a weighted majority vote. Then, TIES only keep the parameter values that aligns with the elected sign, and compute the mean.

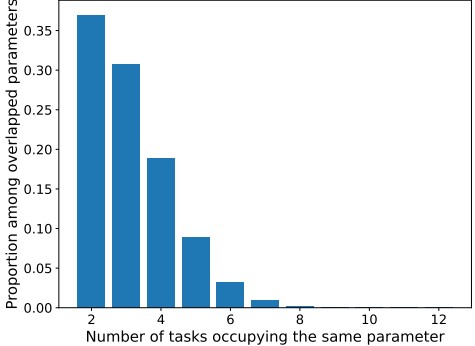

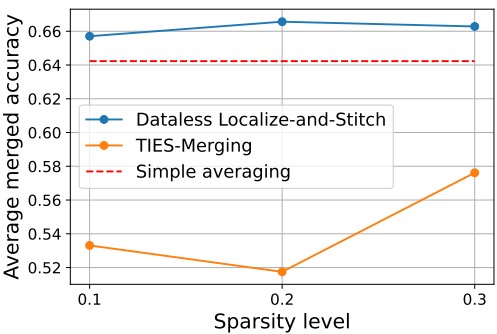

Figure 12: When merging 12 NLP tasks with top-20% selection in TIES, most overlapping regions only involve 2 or 3 tasks. This is the regime where the sign election process in TIES is less effective in.

Figure 13: When dealing with two conflicting tasks, the sign election stage of TIES is not effective. The performance of TIES is consistently worse than simple averaging on all parameters.

This approach by TIES might be more advantageous when overlapping regions involve a larger number of tasks. The rationale is that with more tasks contributing to an overlap, the process of sign determination and selective averaging may more accurately capture the consensus of task vectors for all tasks as a whole. However, when only two tasks are involved (which is often the case as shown in Figure 12), TIES may only retain parameters predominantly from the task with the larger magnitude at each position. In such scenarios, important

parameters for both tasks could be alternately ignored, potentially degrading performance for both tasks. This selective process might, therefore, impair the overall efficacy in maintaining crucial task-specific information, particularly in tightly contested regions. We demonstrate this in Figure 13, where we use the same example of conflicting tasks as in Section 3.1, i.e., QNLI and MNLI. When merging models on two conflicting tasks, the performance of TIES is significantly worse than simple averaging on all model parameters across all sparsity levels.

## D    Comparison between `Localize-and-Stitch` and Consensus Merging

Given the similarity of our approach with Consensus Merging (Wang et al., 2024b), we compare them in detail. In Table 2, we have demonstrated `Localize-and-Stitch` has superior performance, and we provide further empirical analysis as follows.

**Localization area.** Our method is able to localize as little as 1% of total parameters, compared to 20% required by (Wang et al., 2024b). Our small localized regions lead to less task interference, better multi-task performance and much better compression rates. In Consensus Merging, the sparsity is controlled by the hyperparameter $\lambda$, which is chosen among $\{0.2, 0.3, 0.4, 0.5, 0.6\}$. Note that is not directly used as the sparsity, rather, it serves as a scaling factor to determine the magnitude threshold for localization, as shown in Equation (5) in (Wang et al., 2024b). Intuitively, a larger imposes a stricter threshold, resulting in a smaller localized region. We present the resulting average sparsity corresponding to each choice of in Table 14. While the resulting sparsity might vary depending on the specific tasks and models, we find that all choices of $\lambda$ lead to an average sparsity of at least 25%, which is substantially larger than the 1% sparsity achieved by our localization method.

Table 14: Correspondence between $\lambda$ and sparsity.

| $\lambda$ | Average sparsity (language) | Average sparsity (vision) |
|---|---|---|
| 0.2 | 0.4977 | 0.5071 |
| 0.3 | 0.4316 | 0.4144 |
| 0.4 | 0.3864 | 0.3456 |
| 0.5 | 0.3544 | 0.2939 |

**Ablation on sparsity.** Similar to Appendix C, we provide an ablation study to demonstrate that the superior performance of our method is not only due to the sparsity, but also the effectiveness of our localization approach. To achieve the same sparsity level of 1% as our method, we find that we need to set the sparsity hyperparameter $\lambda$ in (Wang et al., 2024b) to be at least 0.95. This value falls significantly outside the range of the hyperparameter space they consider, indicating that TALL masks are unlikely to perform well when used to identify regions this small. Indeed, our experiments in Table 15 confirm that TALL masks with 1% sparsity yield poor performance on both vision and language benchmarks. This further illustrates the effectiveness of our localization approach in identifying tiny regions containing essential information, contributing to reduction of task interference and substantially better compression rates.

Table 15: Comparison with (Wang et al., 2024b) at the same sparsity level.

| Method | Sparsity | Average language acc | Average vision acc |
|---|---|---|---|
| Consensus TA | 1% | 0.477 | 0.518 |
| Consensus TIES | 1% | 0.463 | 0.524 |
| `Dataless Localize-and-Stitch` | 5% | 0.734 | 0.740 |
| `Localize-and-Stitch` | 1% | 0.759 | 0.799 |

**Runtime comparison.** The larger runtime observed (Table 11) for Consensus TA compared to our method is primarily due to its extensive hyperparameter tuning process. Take merging vision models as an example, for the formulation of (Wang et al., 2024b), the hyperparameter tuning process is as follows: i) Tune sparsity for all tasks, which requires 40 evaluations for 5 choices of on 8 tasks. ii) Tune scaling factor , which requires

88 evaluations for 11 choices of on 8 tasks. In total, there are *128 task-specific evaluations required to select the two hyperparameters* following Appendix A.2 in (Wang et al., 2024b), which we find very time-consuming in practice. In contrast, our algorithm does not require hyperparameter tuning. It operates by training on 8-shot data for 10 epochs, a process we find highly efficient.

Furthermore, hyperparameter tuning, particularly for scaling factors, has been shown to greatly influence model merging performance, making methods like Consensus TA sensitive to these choices as highlighted in (Tam et al., 2024). On the other hand, our method avoids these complexities, offering a more efficient and robust approach that enhances performance consistency.

## E  Details on localization

Here, we detail the skill attribution method in Panigrahi et al. (2023) and explain the difference with our formulation. Panigrahi et al. (2023) aims to localize task-specific skills contained in finetuned language models. They introduce model grafting, where for given pretrained and finetuned model parameters $\theta_{\mathrm{pre}}$ and $\theta_{\mathrm{ft}}$, they graft parameters of $\theta_{\mathrm{ft}}$ in the region $\gamma$ onto the pretrained model as

$$\overline{\theta_{\mathrm{ft}}}(\gamma) = \gamma \odot \theta_{\mathrm{ft}} + (1 - \gamma) \odot \theta_{\mathrm{pre}}.$$

With the grafting operation, they find the localized region with the following optimization procedure, where they essentially find the region leading to the best grafted performance.

$$\underset{\gamma \in \{0,1\}^d : \|\gamma\|_o \leq s}{\arg\min} \ell_\tau(\gamma \odot \theta_{\mathrm{ft}} + (1 - \gamma) \odot \theta_{\mathrm{pre}})$$

They also use a reparametrization of the binary mask $\gamma$ as the sigmoid of a real-valued vector $S$, and reformulate the problem as

$$\underset{S \in \mathbb{R}^d}{\arg\min} \ell_i(\gamma \odot \theta_{\mathrm{ft}} + (1 - \gamma) \odot \theta_{\mathrm{pre}}),$$
$$\gamma := \gamma_{base} \odot (1 - \sigma(S)) + (1 - \gamma_{base}) \odot \sigma(S), \tag{3}$$

where $\gamma_{base}$ is the top-$k\%$ largest elements in the task vector. This serves as an initialization for the optimization. In comparison, our formulation is as follows

$$S_i = \underset{S \in \mathbb{R}^d}{\arg\min} \ell_i \left(\theta_{\mathrm{pre}} + \sigma(S) \odot \tau_i\right) + \lambda \|\sigma(S)\|_1,$$

There are two main differences between the formulations. Firstly, our formulation of $S$ is more straightforward, as we directly have $\gamma = \sigma(S)$. In contrast, $S$ in Equation (3) serves as a selector of whether to take the value from $\gamma_{base}$, leading to more complex computation. Secondly, our approach uses the $L_1$ constraint to control the sparsity in a more fine-grained manner, while Equation (3) does not have this constraint, and the authors control the sparsity via early stopping.

Table 16: Accuracy ($^\dagger$F1) comparison of localization methods on RoBERTa-base models on twelve language tasks.

| Task | SST-2 | CR | MR | MPQA | TREC | SUBJ | QNLI | SNLI | MNLI | RTE | MRPC$^\dagger$ | QQP | Average acc/F1 |
|---|---|---|---|---|---|---|---|---|---|---|---|---|---|
| Single-task finetuned | 0.898 | 0.894 | 0.844 | 0.848 | 0.938 | 0.931 | 0.764 | 0.791 | 0.706 | 0.643 | 0.766 | 0.716 | 0.811 |
| **Single-task grafted (Ours)** | 0.897 | 0.883 | 0.855 | 0.844 | 0.918 | 0.933 | 0.751 | 0.772 | 0.703 | 0.639 | 0.745 | 0.708 | **0.804** |
| Single-task grafted (Panigrahi et al., 2023) | 0.902 | 0.908 | 0.862 | 0.851 | 0.884 | 0.925 | 0.752 | 0.756 | 0.676 | 0.643 | 0.757 | 0.693 | 0.801 |
| **Merged (Ours)** | 0.896 | 0.896 | 0.849 | 0.828 | 0.782 | 0.820 | 0.734 | 0.621 | 0.580 | 0.633 | 0.820 | 0.651 | 0.759 |
| Merged (Panigrahi et al., 2023) | 0.897 | 0.895 | 0.847 | 0.831 | 0.816 | 0.803 | 0.727 | 0.649 | 0.580 | 0.633 | 0.819 | 0.656 | **0.763** |

We present the performance comparison of the two localization methods in Tables 16 and 17. In both cases, our approach with Equation (1) outperforms Panigrahi et al. (2023). The performance may come from the fact that Panigrahi et al. (2023) use early stopping to control the sparsity, which results in incomplete optimization for the masks. We also report the merged performance by following the same stitching process. On the language tasks, the performance is similar, while on the vision tasks, our localization leads to better merged performance.

Table 17: Comparison of localization methods on CLIP ViT-B/32 models on eight vision tasks.

| Task | SUN397 | Cars | RESISC45 | EuroSAT | SVHN | GTSRB | MNIST | DTD | Average acc |
|---|---|---|---|---|---|---|---|---|---|
| Single-task finetuned | 0.753 | 0.777 | 0.961 | 0.997 | 0.975 | 0.987 | 0.997 | 0.794 | 0.905 |
| **Single-task grafted (Ours)** | 0.731 | 0.772 | 0.955 | 0.989 | 0.963 | 0.973 | 0.996 | 0.781 | **0.895** |
| Single-task grafted Panigrahi et al. (2023) | 0.731 | 0.750 | 0.935 | 0.959 | 0.929 | 0.932 | 0.976 | 0.753 | 0.871 |
| **Merged (Ours)** | 0.672 | 0.683 | 0.818 | 0.894 | 0.879 | 0.866 | 0.948 | 0.629 | **0.799** |
| Merged Panigrahi et al. (2023) | 0.669 | 0.678 | 0.798 | 0.861 | 0.846 | 0.826 | 0.919 | 0.653 | 0.781 |

## F  GPT2-XL experiment details

For the experiments in Section 4.3, we use the following three checkpoints from Hugging Face:

- `Locutusque/gpt2-large-conversational`

- `Onlydrinkwater/gpt2xl_language_math_520_10base`

- `Rachneet/gpt2-xl-alpaca`

They are all finetuned on the original release of the GPT2-XL model `openai-community/gpt2-xl`. The selection of the three models and associated tasks is a result of an extensive evaluation process. After testing dozens of finetuned GPT-2XL checkpoints, we establish specific criteria to ensure the relevance and rigor of our experiments: The checkpoints should

- be fully finetuned instead of PEFT,

- have a well-defined and evaluable downstream task,

- perform noticeably better than the pretrained model on its respective task.

We find that most finetuned checkpoints do not meet the last criterion, which is crucial for substantiating the benefits of our merging method. Consequently, the three models and tasks combinations chosen best satisfy all three criteria, making them the most appropriate for our purposes.

## G  Datasets

**Vision datasets**  Following the practice in Ilharco et al. (2023), we use the following 8 datasets for the vision part of our experiments:

- SUN397 (Xiao et al., 2016). The Scene UNderstanding dataset contains 108,754 images of 397 classes.

- Stanford Cars Krause et al. (2013). The Stanford Cars dataset contains 16,185 images of 196 classes of cars.

- RESISC45 (Cheng et al., 2017). The REmote Sensing Image Scene Classification dataset contains 31,500 images, covering 45 scene classes.

- EuroSAT (Helber et al., 2019). The EuroSAT dataset consists of 10 classes with 27000 labeled and geo-referenced samples. Each class represents a different land use and land cover.

- SVHN (Netzer et al., 2011). The Street View House Numbers dataset contains 600,000 digit images in 10 classes of printed digits cropped from pictures of house number plates.

- GTRSB (Stallkamp et al., 2011). The German Traffic Sign Recognition Benchmark contains 43 classes of traffic signs with more than 50,000 images.

- MNIST (LeCun et al., 2010). The MNIST dataset contains 60,000 training images and 10,000 testing images of 10 handwritten digits.

- DTD (Cimpoi et al., 2014). The Describable Texture Dataset contains 5,640 texture images in the wild with 47 classes.

SUN397, RESISC45 and DTD are under the Creative Commons Attribution-ShareAlike 4.0 International License. Stanford Cars is under the ImageNet License. EuroSAT is under MIT License. MNIST is under Gnu General Public License. GTRSB and SVHN are under CCBY-SA License.

**Language datasets** Following the practice in Panigrahi et al. (2023), we use the following 12 datasets for the language part of our experiments. The majority comes from the GLUE benchmark (Wang et al., 2018).

- SST-2 (Socher et al., 2013). The Stanford Sentiment Treebank is a sentiment analysis dataset, which contains sentences from movie reviews and human annotated binary sentiments.

- CR (Hu & Liu, 2004). The Customer Review dataset consists of customer reviews on e-commerce products with binary sentiment labels.

- MR (Pang & Lee, 2005). The Movie Review dataset consists of movie reviews with binary sentiment labels.

- MPQA (Wiebe et al., 2005). The Multi-Perspective Question Answering dataset contains news articles and text documents manually annotated for opinions and other private states including beliefs, emotions, sentiments, etc. Here, we use it for binary sentiment classification.

- TREC (Voorhees et al., 1999). The Text REtrieval Conference (TREC) dataset contains 6k questions phrased by users and categorized into a small number of categories. The task is to classify the questions into these categories.

- SUBJ (Pang & Lee, 2004). The SUBJectivity dataset contains 10k movie reviews with an annotation of whether the review describes something subjective or objective about the movie.

- QNLI (Wang et al., 2018). The Question-answering NLI dataset is converted from the Stanford Question Answering Dataset (SQuAD) (Rajpurkar et al., 2016), which contains questions and the paragraphs that contain the answer to the corresponding questions. QNLI converts SQuAD into sentence pair classification by forming a pair between each question and each sentence in the corresponding context, where the task is to predict whether the context contains the answer to the question.

- SNLI (Bowman et al., 2015). The Stanford Natural Language Inference dataset contains 570k sentence pairs manually labeled as entailment, contradiction or neutral.

- MNLI (Williams et al., 2017). The Multi-Genre Natural Language Inference Corpus is a collection of 433k sentence pairs annotated with textual entailment information.

- RTE (Wang et al., 2018). The Recognizing Textual Entialment dataset contains a series of textual entailment challenges, including RTE1 (Dagan et al., 2005), RTE2 (Haim et al., 2006), RTE3 (Giampiccolo et al., 2007) and RTE5 (Bentivogli et al., 2009). The neutral and contradiction classes are combined into a no entailment class.

- MRPC Dolan & Brockett (2005). The Microsoft Research Paraphrase Corpus consists of sentence pairs from online news sources, with human annotations of whether the sentences in the pair are semantically equivalent. Since the classes are imbalanced, we report the F1 score.

- QQP (Iyer et al.). The Quora Question Pairs dataset consists of question-answer pairs from the website Quora. The task is to determine whether two questions are semantically equivalent.

CR, RTE, MRPC, QQP, QNLI are under CCBY-SA License. MRPC is under Microsoft Research License. MNLI is under OANC's License. SNLI is under a Creative Commons Attribution-ShareAlike 4.0 International License.

**GPT datasets**  We introduce the datasets used for GPT2-XL experiments.

- MMLU (Hendrycks et al., 2021). The Massive Multitask Language Understanding measures knowledge in 57 subjects across STEM, humanities, social science, etc.

- ARC (Clark et al., 2018). The AI2 Reasoning Challenge dataset contains 7,787 grade-school level, multiple choice science questions.

- TruthfulQA (Lin et al., 2021). The TruthfulQA dataset measure whether a model is truthful in generating answers to questions. It comprises 817 questions spanning 38 categories, including health, law, finance, etc.

- Alpaca (Taori et al., 2023). The Alpaca dataset contains 52,000 instruction-following examples.

- GSM8k (Cobbe et al., 2021). The Grade School Math 8K dataset contains 8,500 high quality grade school math problems created by human problem writers. The problems take 2 to 8 stpes to solve.

- HotpotQA (Yang et al., 2018). The HotpotQA dataset is a question answering dataset featuring multi-hop questions.

ARC is under CC BY-SA License. TruthfulQA and Alpaca are under Apache License 2.0. MMLU and GSM8k are under MIT License. HotpotQA is under CC BY-SA 4.0 License.

## H   Implementation details

The experiments are run on NVIDIA RTX A6000 GPUs with 48GB memory.

**Finetuning.** For the experiments on RoBERTa-base, we perform the finetuning process following the same procedure as Panigrahi et al. (2023). Specifically, we use a batch size of 4 and a learning rate of 2e-5 to finetune on each of the language tasks for 10 epochs with the SGD optimizer. For the experiments on CLIP ViT, we directly use the finetuned checkpoints provided in Ilharco et al. (2023) with the data preprocessing step provided in (Yang et al., 2023). The finetuned models in the GPT2-XL experiments in provided in Appendix F.

**Localization.** Following the practice in Panigrahi et al. (2023), in the localization step, we initialize the trainable real-valued vector $S$ as the mask for top-$k\%$ largest entries in the task vector. Since the actual map is rounded from $\sigma(S)$ but not $S$, we choose the initial values of $S$ to be either 0 or 3, as $\sigma(3)$ is sufficiently close to 1. To achieve a sparsity level of 1%, we use the learning rate $1e7$, batch size 16, $L_1$ regularization factor $\lambda$ 1e-5 and perform the optimization for 10 epochs on 64-shot data from each task. Following common practice in Panigrahi et al. (2023); Yadav et al. (2023), we only perform localization in the transformer blocks, and do not consider embedding layers.

**Baselines.** We use both task arithmetic and TIES-Merging in a dataless manner, meaning that we directly use their recommended hyperparameters without tuning it. To be specific, for task arithmetic, the recommended scaling factor is 0.4. For TIES-Merging, the recommended scaling factor is 1 and sparsity level is 20%. This ensures a fair comparison with `Dataless Localize-and-Stitch`, which we also apply a fixed sparsity level across all experiments, namely 5%.

