# OpenReview forum: "Localize-and-Stitch: Efficient Model Merging via Sparse Task Arithmetic"
_TMLR — Accepted by TMLR_

### Review · Reviewer_aS31 · 2024-10-15

**Summary Of Contributions:**

The paper introduces "Localize-and-Stitch," an innovative model merging approach that efficiently combines the strengths of multiple fine-tuned models into a unified model while preserving their specialized capabilities. This method addresses the issue of task interference common in global model merging by identifying and merging only essential skill regions within fine-tuned models. Evaluated across various vision and language benchmarks, "Localize-and-Stitch" outperforms existing methods under different data availability scenarios.

**Audience:**

Yes

**Claims And Evidence:**

Yes

**Requested Changes:**

1. Adding a discussion with [1] and comparing it in the experiments, since the method [1] is related to the proposed method in this paper.
2. Adding a discussion with [2], which is about the performance deterioration problem of merged models.

Please refer to Weaknesses for details.

**Strengths And Weaknesses:**

Strengths:

1. This paper proposes a simple yet effective method to merge multiple fine-tuned models into a unified model and the proposed method can be effectively applied in scenarios with or without validation data.
2. Extensive experiments are conducted on vision and language domains to demonstrate the effectiveness of the proposed method.

Weaknesses:
1. The proposed method shares similarities with the approach in [1] that localizes and merges the task-specific information. Thus, it would be beneficial to discuss and compare with it in the paper.
2. Although reducing task conflicts, the performance of merged models using the proposed method is still largely behind that of single-task fine-tuned models, indicating there may be additional challenges in model merging that go beyond task interference. [2] proposes to inject task-specific knowledge into the merged model to alleviate the performance deterioration problem. It would be interesting to discuss with it in the paper.

[1] Wang et al. Localizing Task Information for Improved Model Merging and Compression. ICML, 2024.

[2] Jiang et al. BYOM: Building Your Own Multi-Task Model For Free. arXiv:2310.01886, 2023.

---

> ### Author Response · Authors · 2024-11-02
>
> We want to thank the reviewer for the insightful feedback and the time invested in reviewing our paper. We hope that the following response can address the reviewer’s concerns:
>
> **[Comparison with reference [1]]** We want to clarify several key advantages of our method:
> - **Much smaller localized regions**: Our method is able to localize as little as 1% of total parameters, compared to 20% required by [1]. Our small localized regions lead to less task interference, better multi-task performance and much better compression rates.
> - **Improved performance**: We compare the performance with [1] on vision benchmarks. Our Localize-and-Stitch noticeably outperforms the baselines. Note that [1] requires validation data for tuning merging coefficients, while Dataless Localize-and-Stitch does not require any data.
> - **No need for hyperparameter tuning**: Similar to task arithmetic and TIES, [1] requires tuning merging coefficients, which requires validation data and is time-consuming. In contrast, our method does not involve this hyperparameter (mentioned in Section 3.3), significantly improving efficiency. Following the practice in [1] of searching the merging coefficient over {0.0,0.1,...0.9,1.0}, we report the runtime in the following table, which further demonstrates the efficiency of our method.
>
> |                              | Average Vision performance | Runtime (s) |
> |------------------------------|----------------------------|-------------|
> | Consensus TA [1]             | 0.750                      | 7361.52     |
> | Consensus TIES [1]           | 0.748                      | 26042.43    |
> | Dataless Localize-and-Stitch | 0.740                      | 304.85      |
> | Localize-and-Stitch          | **0.799**                  | 5130.05     |
>
> We will cite this work and include the discussion in our final manuscript.
>
> ---
>
> **[Discussion about reference [2]]** We recognize two main categories of model merging approaches:
> - **Single unified model**: the merged model retains the same architecture and parameter count as the original specialized models (e.g., Localize-and-Stitch).
> - **Mixture of experts**: the merged model incorporates routing mechanisms to direct inputs to task-specific networks (e.g., BYOM).
>
> These approaches present different tradeoffs between storage efficiency and performance. The unified model is storage efficient and incurs no additional overhead during deployment, but its performance may lag behind specialized models. The MoE approach can achieve higher performance by leveraging task-specific knowledge, but requires storing more parameters and, in some cases, additional modules or training. We will cite the referenced work and add the discussion in our final manuscript.

---

### Review · Reviewer_x7BD · 2024-10-15

**Summary Of Contributions:**

The paper addresses the model merging problem, aiming to leverage a pre-trained model across multiple downstream tasks. The proposed method, Localize-and-Stitch, tackles this by first identifying task-specific important but sparse parameters through importance training, and then stitching these parameters together. Extensive experiments across both language and vision tasks demonstrate the approach's effectiveness. Additional ablation studies including visualizations of localized regions and Jaccard Index comparisons, further validate the method's success.

**Audience:**

Yes

**Claims And Evidence:**

Yes

**Requested Changes:**

None

**Strengths And Weaknesses:**

## Strengths
* The motivation behind the proposed method, along with its distinction from prior work, is thoroughly discussed and supported by experiments.
* The method is evaluated across multiple models in both language and vision domains.
* The ablation studies effectively explain the method's success, and additional experiments in the Appendix, such as runtime comparisons, further enhance the completeness of the paper.

## Weaknesses
The reviewer did not find any major weaknesses in the paper. One suggestion is that the proposed method, particularly the localization step, bears a strong resemblance to prior work on model pruning (e.g., [1][2]). It would be valuable to discuss this relationship, as it may reveal opportunities for future work and potential improvements to the method

[1] Sanh et al; Movement Pruning: Adaptive Sparsity by Fine-Tuning.

[2] Zhang et al; PLATON: Pruning Large Transformer Models with Upper Confidence Bound of Weight Importance.
[2]

---

> ### Author Response · Authors · 2024-11-02
>
> We want to thank the reviewer for the insightful feedback and the time invested in reviewing our paper. We hope that the following discussion incorporates the reviewer’s suggestion:
>
> **[Connection with pruning]** Both localization and pruning identify key regions in the parameter space that are important to model performance. The primary difference is that in localization, parameters outside the identified regions are retained at their pretrained values, while in pruning, those parameters are set to be zero, effectively removing them. We will cite both papers and include a discussion in our final manuscript.

---

### Review · Reviewer_Qb5Q · 2024-10-29

**Summary Of Contributions:**

In the pre-training fine-tuning paradigm, model merging has emerged as a promising approach to multi-task learning with the goal of obtaining fine-tuned performance on multiple downstream tasks. However, most existing model merging methods use all the parameters of the pre-trained/fine-tuned model which can lead to task interference.

* As a solution to this problem, the paper proposes a model merging method, `Localize-and-Stitch` as an improved approach.
* The paper first proposes a method to find localized (`localize`) parameters within the fine-tuned neural network which contribute the most to fine-tuned performance. It does this by making a modified formulation of Panigrahi et al. (2023) to train a real-valued vector $S$ over all model parameters to mask the task vector for task $i$, $\tau_i$, with the minimum possible number of parameters.
* The proposed method then combines (`stitch`es) the sparse task vectors into a single model claiming to obtain multi-task performance while minimizing interference at the same time.
* Through experiments on both vision and language benchmarks, the paper shows the benefits of the proposed method and highlights additional benefits.

**Audience:**

Yes

**Claims And Evidence:**

Yes

**Requested Changes:**

Please see the **Weaknesses** section above. I have tried to mention possible experiments and analyses which should make your work more foolproof.

**Strengths And Weaknesses:**

**Strengths**

* The highlighted problem is valid and the proposed method is intuitive, easy to understand and simple to implement making it easily applicable in many scenarios.
* I quite like the way the paper is written. The authors do a good job of highlighting the analysis and the logic behind their formulation of the `Localize-and-Stitch` method.


**Weaknesses**

***Disjointedness of localized parameters across tasks***

* The goal of the work is to avoid task interference. The idea is to localize task vector parameters to a very small subset of parameters.
* While I understand that the objective mentioned in Equation 1 should achieve the sparsity effect in the resulting task vector, the paper lacks a solid analysis of how the generated sparse task vectors are relatively disjoint between tasks.
* Some analysis in this direction is present in Figure 4 of the paper. But

a) how conceptually similar/different are the datasets/tasks analysed in this Figure? Also,

b) how does the cosine similarity heatmap of Figure 4b compare with a similar heatmap from other task vector/model merging methods?

c) In the experiments section, you could create subsets of fine-tuned datasets from Table 4 (say) where some of the dataset subsets have higher conceptual overlap and some have lower. Compare in these 2 settings to highlight the win from this paper.
* In a nutshell, I am trying to say that if avoiding task interference is the goal, the analysis and experiments on the method should highlight that the goal is achieved.


***Model Merging and Continual Learning***

* Model merging and continual learning genres of work have very similar goals, i.e., for a model to be performant at several fine-tuned tasks together. Continual learning does the fine-tuning sequentially (continually) whereas model merging does it often together with multiple individually fine-tuned models.
* I appreciate the paper's highlighted point on a comparison with continual learning methods.
* The paper can also go a step further and:

a) include CL baselines as competitive baselines with the model merging ones it already mentions in Table 5. This would give a bigger picture view on the relative performance benefits/drawbacks of both genres of methods.

b) Specify the pre-trained performance of the final model (merged model I mean) on tasks it wasn't fine-tuned on, highlighting how much of the pre-trained knowledge it forgot as a result of fine-tuning + model merging. Achieving excellent performance on forgetting is a stretch goal for this paper, but nonetheless these insights are important to highlight for the community's benefit.

---

> ### Author Response · Authors · 2024-11-02
>
> We want to thank the reviewer for the insightful feedback and the time invested in reviewing our paper. We hope that the following response can address the reviewer’s concerns:
>
> **Disjointedness of localized parameters across tasks**
>
> **[Analysis of disjoint regions]** Our observation that the localized regions are relatively disjoint comes from the small Jaccard similarity of the overlapping localized regions. The same observation has been reported in [1], suggesting that different tasks often require distinct network components to capture their unique skills. While this intuition is widely accepted, a theoretical understanding of why different tasks are localized to different regions remains an open question in the literature, and is beyond the scope of this paper. We hypothesize that the overlaps between different task vectors will be task-dependent, and need data-dependent assumptions to quantify theoretically.
>
> [1] Task-Specific Skill Localization in Fine-tuned Language Models. Panigrahi et al. 2023.
>
> **[Conceptual similarity of tasks]** In Figure 4, we observe a clear cluster among sentiment classification tasks (SST-2, CR, MR and MPQA), which show notable overlap in their localized regions and high cosine similarity. Another potential cluster is pairwise sentence tasks, including NLI (QNLI, SNLI, MNLI), entailment (RTE, MRPC) and QA (QQP). However, this cluster is less distinct, with relatively low Jaccard similarity, likely due to differences in data sources across these tasks.
>
> **[Heatmap comparison]** The cosine similarities in a heatmap based on full task vectors would be smaller than those observed with our masked task vectors. This is because full task vectors are extremely high-dimensional (the same dimension as the model size). As discussed in the paper, finetuning introduces redundancy, adding randomness to the task vectors. Thus, the full task vectors are likely close to orthogonal, as shown in [2]. In contrast, our localized task vectors, which represent only ~1% of the total parameters, are lower-dimensional and more concentrated in task-specific information. Thus, the combination of Jaccard and cosine similarity in Figure 4 offers a more informative measure of task similarities than would be achievable with full task vectors.
>
> [2] Editing models with task arithmetic. Ilharco et al. 2023.
>
> **[Subsets of datasets]** To assess our method’s effectiveness on tasks with different conceptual overlap, we created two subsets: i) Conceptually similar subset: Composed entirely of sentiment classification tasks (SST-2, CR, MR, MPQA). ii) Conceptually dissimilar subset: Including tasks from different categories (SST-2 for sentiment classification, TREC for question classification, SUBJ for subjectivity, and MNLI for entailment). The following table reports the average performance for each subset. In the similar subset, where tasks share similar skills, all merging methods perform equally well. However, in the dissimilar subset—where task skills differ and may even conflict—Localize-and-Stitch shows a significant advantage, demonstrating its ability to effectively resolve task interference.
>
> |                     | Similar subset | Dissimilar subset |
> |---------------------|----------------|-------------------|
> | Task arithmetic     | 0.878          | 0.802             |
> | TIES                | 0.875          | 0.812             |
> | Localize-and-Stitch | **0.880**          | **0.831**             |

---

> ### Author Response · Authors · 2024-11-02
>
> **Model Merging and Continual Learning**
>
> **[CL baselines]** We include an additional baseline of continual training, where the results are obtained by sequential training on each task, using the model from the previous task as the starting point. While continual training performs better than other model merging baselines, it still falls short of Localize-and-Stitch. This suggests a potential avenue for enhancing continual learning: instead of fine-tuning on top of the last task model, it may be advantageous to first finetune the pretrained model on the new task and then merge it with the existing multi-task model on the old tasks. We leave further exploration of this approach for future work.
>
> |                     | SST-2+CR  | +TREC     | +SUBJ     | +QNLI     | +QQP      |
> |---------------------|-----------|-----------|-----------|-----------|-----------|
> | Task arithmetic     | **0.907** | 0.864     | 0.853     | 0.817     | 0.762     |
> | TIES                | 0.897     | 0.832     | 0.827     | 0.796     | 0.757     |
> | Localize-and-Stitch | 0.906     | **0.905** | **0.897** | **0.856** | **0.827** |
> | Continual training  | 0.897		     | 0.873     | 0.858     | 0.837     | 0.790     |
>
> **[Forgetting of pretrained knowledge]** We have demonstrated this result in Figure 8 of Section 4.4, where the control task is ImageNet classification given its generality as a measure of pretrained knowledge. The pretrained model’s performance is represented by the gray triangle in the lower right corner. Our results show that the merged model using Localize-and-Stitch retains similar ImageNet accuracy to the original pretrained model, noticeably outperforming other merging methods. This demonstrates minimal forgetting achieved by our approach.

---

### Comment · Action_Editor_ojkA · 2024-12-11
**Request for more detailed comparison with the related work [1]**

Given the similarity of the proposed method to [1], a more detailed comparison with it is needed than what was provided in the response to Reviewer aS31. I have the following questions:
- In your response to Reviewer aS31, you claim that the method of [1] requires 20% sparsity. But in [1], the hyper-parameter $\lambda_t$ (which controls sparsity) is tuned for each task, by choosing the best one among a set of choices (see Appendix A.2 therein). Can you clarify your claim?
-  Why does Consensus TA have a larger runtime than your method? The former only requires tuning the scalar hyper-parameters $\lambda_t$ and the merging coefficient $\alpha$, while your method requires optimizing the full task masks.

I would also like to see results on the following additional experiments:
- Compare with [1] on language benchmarks too
- Include an ablation study where you select $\lambda_t$ in [1] to achieve $1%$ sparsity for all task masks, and compare that to your method. This would show if the better performance of your method is only due to higher sparsity.

[1] Wang et al. Localizing Task Information for Improved Model Merging and Compression. ICML, 2024.

---

> ### Author Response · Authors · 2024-12-13
>
> We want to thank the action editor for the insightful feedback and the time invested in reviewing our paper and responses. We hope that the following response can clarify the distinctions of our work from [1]:
>
> ---
>
> **[Sparsity]** From Appendix A.2, the sparsity hyperparameter $\lambda$ is chosen among {0.2,0.3,0.4,0.5,0.6}. Note that $\lambda$ is not directly used as the sparsity, rather, it serves as a scaling factor to determine the magnitude threshold for localization, as shown in Equation (5) in [1]. Intuitively, a larger $\lambda$ imposes a stricter threshold, resulting in a smaller localized region. We present the resulting average sparsity corresponding to each choice of $\lambda$ as follows:
>
> | $\lambda$ | Average sparsity (language) | Average sparsity (vision) |
> |-----------|-----------------------------|---------------------------|
> | 0.2       |                      0.4977 |                    0.5071 |
> | 0.3       |                      0.4316 |                    0.4144 |
> | 0.4       |                      0.3864 |                    0.3456 |
> | 0.5       |                      0.3544 |                    0.2939 |
> | 0.6       |                      0.3309 |                    0.2545 |
>
> While the resulting sparsity might vary depending on the specific tasks and models, we find that all choices of $\lambda$ lead to an average sparsity of at least 25%, which is substantially larger than the 1% sparsity achieved by our localization method. We will make this statement clearer in our final manuscript.
>
> ---
> **[Runtime comparison]** The larger runtime observed for Consensus TA compared to our method is primarily due to its extensive hyperparameter tuning process. Take merging vision models as an example, for the formulation of [1], the hyperparameter tuning process is as follows: i) Tune sparsity $\lambda$ for all tasks, which requires 40 evaluations for 5 choices of $\lambda$ on 8 tasks. ii) Tune scaling factor $\alpha$, which requires 88 evaluations for 11 choices of $\alpha$ on 8 tasks. In total, **there are 128 task-specific evaluations required to select the two hyperparameters** following Appendix A.2 in [1], which we find very time-consuming in practice. In contrast, our algorithm **does not require hyperparameter tuning**. It operates by training on 8-shot data for 10 epochs (detailed in Section 3.2 on page 5), a process we find highly efficient.
>
> Furthermore, hyperparameter tuning, particularly for scaling factors $\alpha$, has been shown to greatly influence model merging performance, making methods like Consensus TA sensitive to these choices as highlighted in [2]. On the other hand, our method avoids these complexities, offering a more efficient and robust approach that enhances performance consistency.
>
> [2] Realistic Evaluation of Model Merging for Compositional Generalization. Tam et al. 2024.
>
> ---
>
> **[Language evaluation]** We conduct the experiments on language benchmarks, and show the results below. Consistent with the vision task results, Localize-and-Stitch outperforms the Consensus merging methods on language benchmarks as well. We will include these results in our final manuscript to provide a comprehensive empirical comparison with [1].
>
> |                              | Average language performance |
> |------------------------------|------------------------------|
> | Consensus TA [1]             |                        0.714 |
> | Consensus TIES [1]           |                        0.695 |
> | Dataless Localize-and-Stitch |                        0.734 |
> | Localize-and-Stitch          |                    **0.759** |
>
> ---
>
> **[Ablation on sparsity]** We conduct an ablation study by setting $\lambda=0$ in the method described in [1]. This configuration ensures no entries are discarded from the task vectors, effectively achieving a sparsity level of 1, or full utilization of the parameters.
>
> We report the results for both the vision and language tasks as follows. In general, using all parameters in the task vectors during merging performs worse than merging localized task vectors.
> As stated in Section 3.1, the superior performance of our method is not solely due to higher sparsity, but also due to the effectiveness of localizing key informative regions for each task. As shown in the results below, our approach still outperforms [1].
>
> |                              | Average language performance | Average vision performance |
> |------------------------------|------------------------------|----------------------------|
> | Consensus TA (sparsity=1)    | 0.704                        | 0.702                      |
> | Consensus TIES (sparsity=1)  | 0.691                        | 0.715                      |
> | Dataless Localize-and-Stitch | 0.734                        | 0.740                      |
> | Localize-and-Stitch          | 0.759                        | 0.799                      |

---

> ### Comment · Action_Editor_ojkA · 2024-12-13
>
> Thank you for the clarifications and the additional experiments. For the ablation study, I meant set $\lambda$ to achieve $1\\%$ sparsity, to compare both methods under the same sparsity. Sorry for the typo!

---

> > ### Author Response · Authors · 2024-12-14
> >
> > Thank you for the clarification. To achieve a sparsity level of 1%, we find that we need to set $\lambda\approx 0.95$. This value falls significantly outside the range of the hyperparameter space the authors consider, indicating that TALL masks are unlikely to perform well when used to identify regions this small. Indeed, our experiments confirm that TALL masks with 1% sparsity yield poor performance on both benchmarks. This further illustrates the effectiveness of our localization approach in identifying tiny regions containing essential information, contributing to reduction of task interference and substantially better compression rates.
> >
> > |                              | Sparsity | Average language performance | Average vision performance |
> > |------------------------------|----------|------------------------------|----------------------------|
> > | Consensus TA                 | 1%       | 0.477                        | 0.518                      |
> > | Consensus TIES               | 1%       | 0.463                        | 0.524                      |
> > | Dataless Localize-and-Stitch | 5%       | 0.734                        | 0.740                      |
> > | Localize-and-Stitch          | 1%       | 0.759                        | 0.799                      |

---

### Comment · Action_Editor_ojkA · 2024-12-21
**Changes needed for camera-ready**

Thank you for submitting the camera-ready version, and implementing most of the requested changes. Some revisions are still needed:
- The authors' names are not formatted according to the TMLR stylefile instructions. Please check the instructions and fix that.
- Run a spell checker on the paper. I noticed several typos for example in Appendix D, such as: "Localizaiton"
- The description of (Wang et al., 2024b) in Table 1 is not clear: $\tau_{MTL}$ is not defined, and "Optimize for TALL masks" is confusing; there's no optimization here, and the "TALL" acronym was not defined. I suggest using instead "Construct task masks" as description for that step, and include the definition of $\tau_{MTL}$. It should also be clarified that this method can be combined with different model merging methods, which can be used to construct $\tau_{MTL}$.
- Include (Wang et al., 2024b) as a baseline in Tables 3 and 4 too.
- Add the performance metric to Tables 5 & 6, and Figures 2, 3, 6 & 7.
- To account for the variance in task difficulties, in addition to the absolute average accuracy, report the performance in terms average *normalized* accuracy / F1 score as done in (Wang et al., 2024b); see Appendix A therein. This is particularly important in cases where you're using different metrics (e.g., accuracy/F1 score for NLP tasks), so simply averaging the absolute values does not make sense.

---

> ### Author Response · Authors · 2025-01-03
>
> Dear action editor,
>
> Thank you for the followup and we have revised each point carefully. We also change the presentation of the continual learning results from table (originally Table 4) to figure (now Figure 9) for better illustration.
>
> Best regards,
>
> Authors

---

> > ### Comment · Action_Editor_ojkA · 2025-01-03
> >
> > Thank for making the requested revisions. Some revisions are still needed:
> > - The definition of $\tau_{MTL}$ in Table 1 is still not clear. I suggest using "where $\tau_{MTL} = \theta_{merged} - \theta_{pre}$,  with $\theta_{merged}$ obtained using any model merging method".
> > - Please specify in all tables including the dataset MRPC that this is the dataset for which you use F1 score. Take a look at the tables in (Panigrahi et al., 2023) for an example on how to do this concisely.
> > - Include the normalized accuracy in Table 4 too.

---

> ### Author Response · Authors · 2025-01-04
>
> Dear action editor,
>
> Thank you again for your followup. We have carefully addressed the first two points. We put the indicator for MRPC whenever possible, and included the information in the caption of Table 2, which does not explicitly contain MRPC.
>
> For the third point, we have considered the inclusion of normalized accuracy in Table 4. However, given the complexity and density of information already present in this table, adding normalized accuracy would overly complicate the visualization and interpretation of the data. Similar works reporting per-task performance do not contain normalized accuracy either, such as Table 8-14 in [1] and Table 1-4 in [2]. In addition, the insights provided by normalized accuracy can be readily discerned from data presented in Table 2.
>
> [1] TIES-MERGING: Resolving Interference When Merging Models. Yadav et al. 2023.
>
> [2] AdaMerging: Adaptive Model Merging for Multi-Task Learning. Yang et al. 2024.

---

### Decision · Action_Editor_ojkA · 2024-12-16

**Recommendation:** Accept with minor revision

**Comment:**

The paper considers the problem of merging multiple models, fine-tuned from the same pretrained model, each on a specific downstream task. The goal is to efficiently merge the fine-tuned models into a single multi-task model that will perform well on all the tasks.
Most existing model merging methods merge models globally across all parameters, which can lead to task interference.
The authors propose a new model merging approach called Localize-and-Stitch to address this issue. The proposed method identifies in each fine-tuned model, a small (1% of the total parameters) localized region  important for the corresponding downstream task, then adds the average of these localized regions back into the pretrained model. The localization method used is based on that of [Panigrahi et al., 2023], with some modifications which improves over it. Experiments show superior performance over existing methods, on both vision and language benchmarks, under different data availability scenarios.

Strengths:
- The paper is well written, with a clear motivation, and presentation of the proposed method and its relation to prior work.
- The problem studied is important
- Proposed method is simple to implement and outperforms existing model merging method. Choosing very sparse regions from each fine-tuned model also leads to advantages beyond multi-task performance, namely better model compression, task interpretability, and preservation of pretrained knowledge.
- Claims are well supported by experiments on both vision and language benchmarks, and ablation studies effectively explain the method's success.

Weaknesses:
- the performance of merged models using the proposed method is still behind that of single-task fine-tuned models
- a more detailed comparison with the related work [1] is needed.

All reviewers recommended to accept the paper and weakly recommended it for the ICLR Journal-to-Conference Track. The authors have addressed in their responses the reviewers' concerns. My main concern was about the big similarity between the proposed method and that of [1]. The authors provided some experimental results showing that their method outperforms that of [1].
Hence, I am recommending to accept this paper with minor revision.

[1] Wang et al. Localizing Task Information for Improved Model Merging and Compression. ICML, 2024.


Requested revisions:
- add the results on tasks with different conceptual overlap and the results of the additional baseline of continual training, from the response to Reviewer Qb5Q.
- comparison with the method in [1]:
	- Discuss the difference between your method and [1] (beyond just the advantages of your method highlighted in the response to Reviewer aS31)
	- Add the method of [1] as a baseline in all the experiments in the paper.
	- Include the ablation study you provided where you select $\lambda_t$ in [1] to achieve $1\\%$ sparsity for all task masks
	- Include the explanation you provided on why Consensus TA has a larger runtime than your method.
- Include the discussion about reference [2] from the response to Reviewer aS31.
- Specify the performance metric in all the results tables and figures.
- Add a discussion of the relation between localization and pruning (beyond the comment in your response to Reviewer x7BD). Note that most pruning methods can be easily adapted to the localization setting.

- Add a discussion of related work on merging models trained on the same task with different training configurations, such as (and many others):
	- Sidak Pal Singh and Martin Jaggi. Model fusion via optimal transport. Advances in Neural Information Processing Systems, 2020.
	- Samuel Ainsworth, Jonathan Hayase, and Siddhartha Srinivasa. Git Re-Basin: Merging Models modulo
Permutation Symmetries. In International Conference on Learning Representations, 2023.
	- Jolicoeur-Martineau, A., Gervais, E., Fatras, K., Zhang, Y., & Lacoste-Julien, S. (2023). Population parameter averaging (papa). arXiv preprint arXiv:2304.03094.

Suggested optional revisions that would strengthen the paper:
- As stated in [1], since their method operates on the multi-task vector, it can be combined with any model merging method. It would be interesting to include the combination of your method with that of [1] as another baseline in your experiments.
- Compare your localization method with some SOTA pruning methods (beyond the very simple top-k baseline) adapted to localization setting. This has the potential of improving your method!

**Audience:**

Yes

**Claims And Evidence:**

Yes